# DPM: Dual Preferences-based Multi-Agent Reinforcement Learning

## Abstract

Preference-based Reinforcement Learning (PbRL), which optimizes reward functions using preference feedback, is a promising approach for environments where handcrafted reward modeling is challenging. Especially in sparse-reward environments, feedback-based reward modeling achieves notable performance gains by transforming sparse feedback signals into dense ones. However, most PbRL research has primarily focused on single-agent environments, with limited attention to multi-agent environments. In this paper, we propose Dual Preferences-based Multi-Agent Reinforcement Learning (DPM), which extends PbRL to multi-agent tasks by introducing *dual* preferences comparing not only whole trajectories but also individual agent contributions during transitions. Furthermore, DPM replaces human preferences with those generated by LLMs to train the reward functions. Experimental results in the StarCraft Multi-Agent Challenge (SMAC) and SMACv2 environments demonstrate significant performance improvements over baselines, indicating the efficacy of DPM in optimizing individual reward functions and enhancing performances in sparse reward settings.

## 1 Introduction

Cooperative multi-agent reinforcement learning (MARL) has demonstrated strong performance across various domains (Du & Ding, 2021; Oroojlooy & Hajinezhad, 2023). However, reinforcement learning-based decision-making methodologies have limited performance in environments where the reward signals are rarely given hence learning the optimal policy is challenging. The situation worsens in multi-agent scenarios since exploration is much harder due to larger state and action space, and each agent's behavior can be assessed only with a shared reward signal.

Preference-based Reinforcement Learning (PbRL) is one notable approach to addressing sparse reward challenges. By training a reward model based on human preferences, PbRL can transform a sparse reward environment into a dense reward environment, thereby allowing for the facile resolution of issues arising from sparse rewards. Recent works have demonstrated that PbRL effectively solves single-agent reinforcement learning tasks in sparse reward setting or even without rewards from the environments, proving PbRL to be an effective alternative (Lee et al., 2021a; Kim et al., 2023).

However, a challenge in applying PbRL to MARL arises from the limitation in optimizing the reward function and its application in MARL has been explored in only a few studies (Zhu et al., 2024). Common methods that rely on a single preference type comparing trajectory pairs struggle to accurately assess the contributions of individual agents, making it difficult to optimize the reward functions. In particular, trajectory preferences pose the challenging *temporal credit assignment problem*, where it is difficult to identify states or actions within a trajectory that influence the reward (Wirth et al., 2017). This issue is further exacerbated by the expansive space in MARL. For example, as depicted in Figure 1 (Left), even within a single trajectory, cooperative and non-cooperative behaviors are mixed, so identifying non-cooperative actions through trajectory comparison alone becomes challenging.

In this paper, we present the **D**ual **P**references-based **M**ulti-Agent Reinforcement Learning (DPM), a model that utilizes Preference-based Reinforcement Learning (PbRL) to tackle the sparse reward issue in Multi-Agent Reinforcement Learning (MARL). DPM provides agent-specific reward information by facilitating comparisons not only between trajectories but also across agents. By integrating preferences that assess the contributions of individual agents, DPM enhances the optimization of

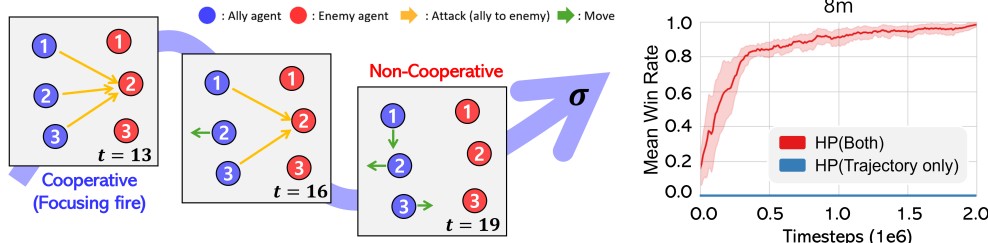

Figure 1: **Motivating Examples** (Left) An example of trajectory selected for preference feedback in the SMAC 3m scenario. Within a trajectory, there are both cooperative($t = 13$) and non-cooperative($t = 19$) scenes, and even within a single scene($t = 16$), there are agents exhibiting cooperative behavior and those that do not. This makes it difficult to train the reward models based solely on trajectory comparisons. (Right) Comparison of win rates in the SMAC 8m scenario based on preference feedback type using human preferences. The red line indicates the use of both types of feedback, while the blue line(with a win rate of always zero) represents the use of only trajectory comparison feedback.

reward functions. As shown by the red line in Figure 1 (Right), employing two types of preferences leads to a higher mean win rate than using a single type.

For DPM's implementation, we use AI-generated feedback from a Large Language Model (LLM), which is recognized for its human-level comprehension (Bai et al., 2022; Lee et al., 2023), instead of human preference feedback. This approach enables smoother training and minimizes the risk of human error.

The experiments are conducted in the sparse reward settings of SMAC and SMACv2 (Ellis et al., 2024) environment. Our proposed model brings significant performance improvements across various scenarios compared to existing MARL baselines. Furthermore, compared to the cases which rely solely on trajectory comparisons, our method demonstrates more stable convergence and higher win rates, indicating better optimization of individual reward functions through dual preference types.

## 2 BACKGROUND

### 2.1 A COOPERATIVE MULTI-AGENT REINFORCEMENT LEARNING

A cooperative MARL task can be formulated as a Dec-POMDP (Oliehoek et al., 2016) which consists of a tuple $\langle S, A, P, R, O, \Omega, n, \gamma \rangle$. $s \in S$ is the global environment state. At each time step, each agent $i \in N \equiv \{1, ..., n\}$ obtains an observation $o^i \in O$ with the observation function $\Omega(s, i) : S \times N \to O$, and selects an action $a^i \in A$ which forms a joint action $\mathbf{a} = \{a^1, \cdots, a^n\} \in A^n$. Then the environment follows the transition function $P(s'|s, \mathbf{a}) : S \times A^n \times S \to [0, 1]$ and all the agents share the same reward function $r(s, \mathbf{a}) : S \times A^n \to \mathbb{R}$. The objective is to learn a joint policy $\pi$ to maximize the expected return $\mathbb{E}_{s_{t+1:\infty}, \mathbf{a}_{t+1:\infty} \sim \pi}[\sum_{k=0}^{\infty} \gamma^k r_{t+k}|s_t, \mathbf{a}_t]$ with $\gamma \in [0, 1)$.

### 2.2 SPARSE REWARD PROBLEM AND SOLUTIONS IN REINFORCEMENT LEARNING

In sparse-reward setting, non-zero rewards $r(s, \mathbf{a})$ are rarely given (e.g., when the given task is completed). To address the sparse-reward challenge, various approaches have been proposed. A common solution is reward shaping in which subgoal-based methods adopt a hierarchical architecture to decompose the given task to smaller sub-tasks (Tang et al., 2018; Jeon et al., 2022). Influence-based methods quantify the influences caused by each agent's action to choose optimal actions (Jaques et al., 2019; Li et al., 2022). DPM instead utilizes preference data to address sparsity. Intrinsic motivation for exploration gives incentives for visiting diverse environmental states (Gronauer & Diepold, 2022). DPM assigns intrinsic rewards for choosing preferable actions.

There are various attempts to solve the sparse reward problem using LLMs. One proposed method leverages the coding abilities of LLMs to design reward functions (Xie et al., 2023), while another involves independently generating multiple reward functions with LLMs and selecting the best-performing one (Ma et al., 2023). Moreover, Sun et al. (2024) enhances the performance of the LLM coder by adding an evaluator that generates preference feedback, resulting in the creation of a robust reward model. The method for identifying key states involves using LLMs to extract critical states, enabling effective exploration of the expansive state-action spaces in MARL (Qu et al., 2024). DPM shares the aspect of using LLMs, but it differs in that LLMs are used to obtain preferences rather than to extract key states or generate code for reward models.

### 2.3 Preference-based Reinforcement Learning

Preference-based Reinforcement Learning (PbRL) is an alternative approach for complex tasks where designing a suitable reward function is difficult. In PbRL, the agent's learning is also guided by a preference between difference behaviors rather than just a single scalar feedback from the environment. The source of preferences could be human feedback (Christiano et al., 2017; Casper et al., 2023; Lee et al., 2021b;a), a scripted teacher which assigns preferences according to true task rewards or AI feedback such as that generated by Large Language Models (Bai et al., 2022; Lee et al., 2023; Klissarov et al., 2023).

A common approach for PbRL is to assign preferences over two trajectory segments (Christiano et al., 2017). A segment $\sigma$ is a sequence of observations and actions during $k$ timesteps $\{s_t, \mathbf{a}_t, \cdots, s_{t+k-1}, \mathbf{a}_{t+k-1}\}$ in single-RL, and we generate preference labels $y \in \{0, 0.5, 1\}$ for each segment pair $(\sigma^1, \sigma^2)$ where $y = 0$ and $y = 1$ mean $\sigma^1$ and $\sigma^2$ is preferred, respectively, and $y = 0.5$ implies both segments are equally preferable. Following the Bradley-Terry model (Bradley & Terry, 1952), the probability of the preference can be defined as:

$$P_\psi[\sigma^1 \succ \sigma^2] = \frac{\exp(\sum_t \hat{r}_\psi(s_t^1, \mathbf{a}_t^1))}{\sum_{i \in \{1,2\}} \exp(\sum_t \hat{r}_\psi(s_t^i, \mathbf{a}_t^i))} \tag{1}$$

where $\sigma^1 \succ \sigma^2$ indicates $\sigma^1$ is preferred to $\sigma^2$, and $\hat{r}_\psi$ is a reward function from preferences and $\psi$ refers to the learnable weights of the reward function. Given the preference dataset $\mathcal{D} = \{(\sigma^1, \sigma^2, y)\}$, the loss of $\hat{r}_\psi$ is the negative log-likelihood and is as follows:

$$\mathcal{L}(\hat{r}_\psi) = -\mathbb{E}_{(\sigma^1, \sigma^2, y) \sim \mathcal{D}} \Big[ (1 - y) \log P_\psi[\sigma^1 \succ \sigma^2] + y \log P_\psi[\sigma^2 \succ \sigma^1] \Big] \tag{2}$$

## 3 Method: DPM

In this section, we present Dual Preferences-based Multi-Agent Reinforcement Learning (DPM), which applies preference-based learning to multi-agent systems based on dual type preferences. DPM not only offers a solution to the sparse reward problem but also replaces human preferences with large language model-based preferences, thereby addressing the issues associated with human preferences. DPM is based on an off-policy online learning MARL algorithm such as QMIX (Rashid et al., 2018).

### 3.1 Overview

The overall structure of DPM is illustrated in Figure 2. DPM comprises reward models, which generate intrinsic rewards, learned from preference feedback. Transition data from the environment and intrinsic rewards are stored in the replay buffer and utilized in the policy training. DPM trains the reward models based on two types of preference feedback. One involves comparing trajectory pairs, while the other entails ranking the actions of agents in a scene. Preferences are obtained using an LLM. To utilize an LLM, vector-based transition information must be transformed into text-based prompts. Therefore, a prompt generator is utilized to convert transition information into text format for input into the LLM. The LLM utilizes the provided information to generate preferences or rankings. The generated preferences (or rankings) are used to train the reward models.

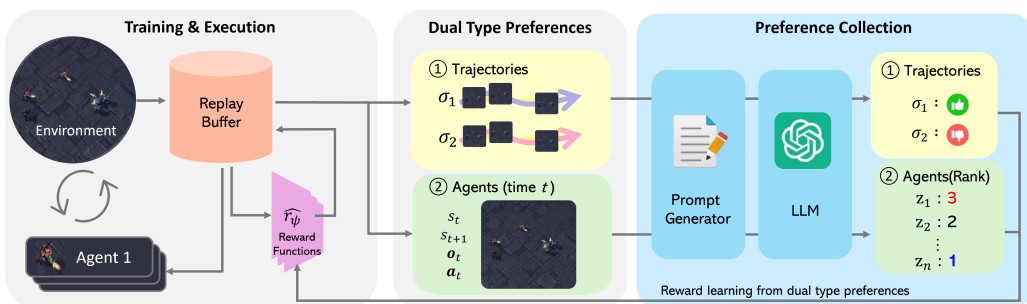

Figure 2: Overview of DPM framework. **Dual Type Preferences**: DPM uses two types of preference feedback. The first is a preference comparing trajectory pairs, and the second is a preference(ranking) comparing the contributions of the agents' actions. **Preference Collection**: DPM obtains preference feedback from an LLM. The prompt generator converts vector-form states and actions into text-based prompts, and the LLM generates preference feedback based on the prompts. **Training and Execution**: The intrinsic reward models are trained based on the preference feedback, and the policy is trained using the rewards generated by the reward models.

## 3.2 DUAL TYPE PREFERENCES

DPM utilizes two types of preferences to train the reward models. One is trajectory comparison preference feedback, which selects the better trajectory through comparison. Trajectory comparison is similar to a common approach in PbRL and consistent with Section 2.3. The other is agent comparison preference feedback, which ranks the actions of agents in a single step. In MARL, since agents have different trajectories even within the same episode, comparing entire trajectories at a macro level cannot provide appropriate rewards for each agent's actions. To resolve this, it is necessary to evaluate at a micro level whether the actions of agents at each step are appropriate. To ensure proper evaluation at the micro level, the actions of agents at a step are ranked, and the rankings are used as preferences.

**Trajectory Comparison**: In a multi-agent concept, a trajectory segment includes observations compared to a single RL segment $\sigma = \{(s_0, \mathbf{o}_0, \mathbf{a}_0), \cdots, (s_k, \mathbf{o}_k, \mathbf{a}_k)\}$. Two trajectory segments are sampled from the replay buffer to generate a preference label ($y$) and we save the pair and the label into the dataset $\mathcal{D}_T = \{(\sigma^1, \sigma^2, y)\}$.

**Agent Comparison**: Agent comparison involves preferences in the form of rankings. In a given step, the actions of agents $(a_t^1, a_t^2, \ldots, a_t^n)$ are ranked according to their contributions. Therefore, when the state $s_t$ and the actions $\mathbf{a}_t$ are provided, the LLM generates ranking labels $\mathbf{z} = \{z_1, z_2, \ldots, z_n\}$ based on the contributions and we save the dataset $\mathcal{D}_A = \{(s_t, \mathbf{a}_t, \mathbf{o}_t, \mathbf{z})\}$ to the buffer.

## 3.3 PREFERENCE COLLECTION

**Trajectory Selection Strategy**: Selecting appropriate trajectories for preference feedback is crucial. To achieve this, common PbRL research (Lee et al., 2021b) has used ensemble-based sampling techniques. In DPM, since the contributions of agents can be ranked, we select trajectories (or scenes) based on the agreement of rankings assigned by the reward functions using Kendall's Tau (Kendall, 1938). Kendall's Tau is used to assess the consistency of ranking data generated by reward models when they create rewards for one step and then rank them based on the generated rewards. A low Kendall's Tau value indicates a disagreement among reward models, suggesting that the reward models are encountering unseen trajectory. Therefore, DPM picks the trajectory with a low average Kendall's Tau value across its steps. For detailed explanation, refer to the Appendix C.

**Prompt Generation**: To obtain preference feedback using a Large Language Model (LLM), prompt generation is essential. However, most environments provide information in the form of vectors or images rather than text. We use a prompt generator to convert vector-based states into text-based prompts that an LLM can understand. The prompt generator employs a template-based approach, where the provided information is substituted into the corresponding sections of the

template. Specifically, A predefined prompt contains placeholders and the placeholders are filled with the corresponding information from the state or observation. The prompt generator effectively converts vector data into text format. However, it is limited in its ability to include all transitions of trajectories in the prompt. Therefore, for trajectory comparison, the prompt only includes the information of the initial state and the end state. Prompt examples can be found in the Appendix E.

**LLM Choice**: The LLM generates preferences for the given comparison dataset using the prompts created by the prompt generator. We utilize the GPT-3.5 and GPT-4o (Achiam et al., 2023) as the preference generation model. This model is considered to possess human-level judgment, enabling it to make decisions at a level comparable to that of humans (Bai et al., 2022; Lee et al., 2023),

### 3.4 REWARD MODELS

**Structure**: In contrast to common MARL approaches that utilize a global (team) reward function, DPM generates rewards individually for each agent by leveraging preferences based on agent comparisons. These reward functions take as input the transition and state information of the agent $(s_t, s_{t+1}, o_t^i, o_{t+1}^i, a_t^i)$ and produce corresponding an intrinsic reward ($\hat{r}_t^i$). This reward generation process enables DPM to tailor rewards to the specific contributions of each agent, enhancing its effectiveness in multi-agent environments. For more details of the reward models' structure, please refer to Appendix A.4. Furthermore, DPM adopts multiple reward models, then the intrinsic reward is defined as the average of rewards generated by the reward models.

**Reward Model Training**: DPM trains the reward models using two types of preference feedback; therefore, the loss function must consider the losses arising from both types. In detail, the loss function reflects both the loss for trajectory comparison feedback($L^T$) and the loss for agent comparison feedback($L^A$), and is as follows :

$$\mathcal{L}(\hat{r}_\psi) = \mathcal{L}^T + \mathcal{L}^A. \tag{3}$$

Both of these losses are defined based on the Bradley-Terry model and the cross-entropy, similarly to common preference-based learning. Specifically, the probability model for the trajectory comparison feedback is defined as follows:

$$P_\psi[\sigma^1 \succ \sigma^2] = \frac{\exp(\sum_t \hat{R}_t^1)}{\sum_{i \in \{1,2\}} \exp(\sum_t \hat{R}_t^i)}, \tag{4}$$

where $\hat{R}_t^i := \sum_{j \in N} \hat{r}_\psi(s_t, a_t^j, o_t^j)$ represents the sum of individual rewards at time $t$ in trajectory segment $i$. Then, the corresponding loss function is

$$\mathcal{L}^T = -\mathbb{E}_{(\sigma^1, \sigma^2, y) \sim \mathcal{D}_T} \Big[ (1-y) \log P_\psi[\sigma^1 \succ \sigma^2] + y \log P_\psi[\sigma^2 \succ \sigma^1] \Big]. \tag{5}$$

The loss for agent comparison feedback at a single step is similar to the approach used to calculate action preferences in (Wirth et al., 2017). We define the loss function by breaking down the complete ordering of agent preferences into a series of pairwise comparisons, as follows:

$$\mathcal{L}^A = -\mathbb{E}_{(s_t, \mathbf{a}_t, \mathbf{o}_t, z) \sim \mathcal{D}_A} \Big[ \frac{1}{|M|} \sum_{(i,j) \in M} \beta_{i \succ j} \log P_\psi[a_t^i \succ a_t^j] \Big], \tag{6}$$

where $M$ is the set of agent pairs, $M = \{(i,j) | i, j \in N, i \neq j\}$, $|M|$ denotes the total number of pairs $(i,j)$ in $M$, and

$$\beta_{i \succ j} := \begin{cases} 0 & z_i > z_j \\ 1 & z_i < z_j \\ 0.5 & z_i = z_j \end{cases} \tag{7}$$

which indicates the observed LLM-generated preference for choosing $a_t^i$ over $a_t^j$. Here, we follow the Bradley-Terry choice model for the pair comparisons as follows:

$$P_\psi[a_t^i \succ a_t^j] = \frac{\exp(\hat{r}_\psi(s_t, a_t^i, o_t^i))}{\sum_{k \in \{i,j\}} \exp(\hat{r}_\psi(s_t, a_t^k, o_t^k))}. \tag{8}$$

## 4 EXPERIMENTS

### 4.1 SETUP

**Environment and Baselines**: We evaluate DPM on StarCraft Multi-agent Challenge (SMAC) (Samvelyan et al., 2019) which consists of diverse micro control task and is one of the most widely used benchmarks for MARL, and SMACv2 which addresses the deterministic limitations of SMAC (Ellis et al., 2024). For baselines, We compare DPM with the common MARL algorithms including VDN (Sunehag et al., 2017), QPLEX (Wang et al., 2020) and Finetuned-QMIX (Hu et al., 2021) which builds upon QMIX (Rashid et al., 2018) incorporating hyper-parameter optimization and other enhancements to achieve state-of-the-art performance. Furthermore, we also test DPM against MASER (Jeon et al., 2022), FOX (Jo et al., 2024) and ICES (Li et al., 2024) which addresses sparse-reward cooperative tasks.

We conduct experiments by setting both the SMAC and SMACv2 environments to sparse reward conditions. In previous researches (Jeon et al., 2022; Jo et al., 2024; Li et al., 2024), the sparse reward setting is applied in a less weak manner such as weak sparse setting in Table 1. However, the setting is not entirely sparse and provides more cues that make it easier for the agents to solve the problem. Therefore, to create a fully sparse reward environment, we adopt a reward setting that provides rewards only upon victory like strong sparse setting in Table 1. We report the average win rates with the standard deviation from three different random seeds. Further details on the experimental setup can be found in Appendix A.

**MARL Algorithms for DPM and Training**: We adopt the Finetuned-QMIX algorithm as our baseline for agent training. This algorithm performs well in dense reward environments, where rewards are provided frequently, as illustrated in Table 1, enabling the agent to effectively learn from the reward signals. However, its performance diminishes in sparse reward settings. To demonstrate that the intrinsic rewards generated by DPM can effectively substitute sparse rewards with dense ones, we also compare DPM with QMIX in dense reward settings. For DPM implementation, we integrate intrinsic rewards ($\hat{r}$) alongside the extrinsic global reward ($r^{ext}$) provided by the environment. The total reward ($r$) used for agent training is then defined as : $r_t = \hat{R}_t + r_t^{ext}$ where $\hat{R}_t := \sum_{i \in N} \hat{r}_\psi(s_t, a_t^i, o_t^i)$ refers to the sum of individual intrinsic rewards at time $t$.

### 4.2 MAIN RESULTS

In this subsection, we conduct experiments in the sparse reward setting of SMAC to evaluate whether DPM can overcome the sparse reward environment. The results are presented in Figure 3.

**Sparse to Dense**: QMIX(Dense) represents the results of the QMIX algorithm operating under the dense reward setting in Table 1. Even though DPM operates in the sparse reward setting, it achieves performance comparable to QMIX(Dense) in SMAC and SMACv2 scenarios. This demonstrates that DPM can transform a sparse reward environment into one similar to a dense reward setting through its intrinsic reward functions. To elaborate, the dense reward setting is manually crafted by human experts. DPM's ability to achieve results on par with those in this setting suggests that it can automatically generate an appropriate reward function using preferences, effectively matching the level of expert-designed reward modeling.

**Overall Performance**: Across eight scenarios, DPM outperformed the baseline algorithms. In the EASY scenarios(3m, 2m_vs_1z, 3s_vs_3z, 2s_vs_1sc, 8m, and 2s3z), DPM achieves an almost 100% win rate. This is a significant performance improvement compared to dense reward-based

Table 1: Rewards according to the reward settings

|  | Dense reward | Weak sparse reward | Strong sparse reward |
|---|---|---|---|
| All enemies die (Win) | +200 | +200 | +200 |
| One enemy dies | +10 | +10 | - |
| One ally dies | +10 | -5 | - |
| Enemy's health | -Enemy's remaining health | - | - |
| Ally's health | +Ally's remaining health | - | - |
| Other elements | +/-with other components | - | - |

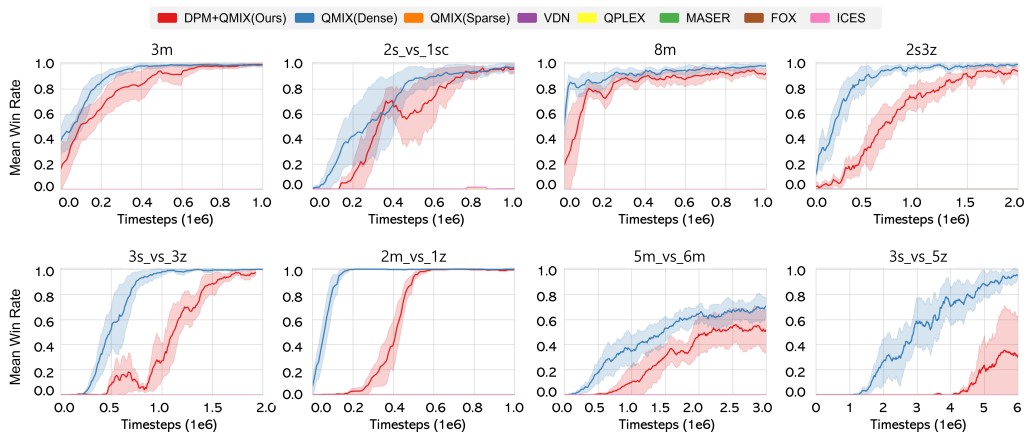

Figure 3: Comparison of performance between DPM and baselines in the sparse reward setting of SMAC. The results show that DPM (red line) outperforms all the baselines, and reaches comparable performances to dense setting (blue line) in several scenarios.

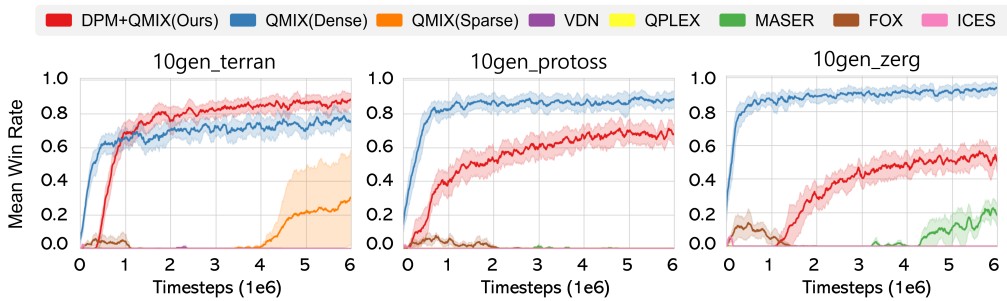

Figure 4: Comparison of performance between DPM and baselines in the sparse reward setting of SMACv2. In all scenarios, DPM outperforms the baselines and is also comparable to cases using dense rewards.

algorithms such as QMIX, VDN, and QPLEX, which record a 0% win rate in the sparse reward setting. The `5m_vs_6m` and `3s_vs_5z` scenarios are categorized as `HARD` scenarios in SMAC, posing significant challenges even for algorithms designed to address sparse rewards. However, DPM outperforms baseline algorithms in these scenarios, demonstrating its robustness.

Additionally, when compared to spare reward-bases algorithms such as MASER, FOX and ICES, DPM shows superior performance. The reason why sparse-reward based algorithms do not perform well is that the experiments are conducted in a harsher sparse setting compared to the previous experimental environment. In contrast, DPM performs well even in the harsher setting.

Figure 4 compares the performance of DPM and baselines in SMACv2. In all three scenarios, DPM significantly outperforms both dense-reward based algorithms and spare-reward based algorithms. Specifically, in the `10gen_terran` scenario, DPM achieves a higher win rate than QMIX in the dense reward setting(QMIX(Dense)). This demonstrates that DPM can perform well even in stochastic environments.

**Scalability and Heterogeneous**: DPM performs well even when there are many agents, such as in the `8m` scenario. In cases where the number of agents is large, DPM can be extended by comparing only a subset of agents rather than all of them. Detailed explanation can be found in the Section 4.5. Moreover, DPM performs well in scenarios where the agents are heterogeneous, such as in the `2s3z` scenario. This indicates that DPM is not only scalable but also generally applicable across various environments.

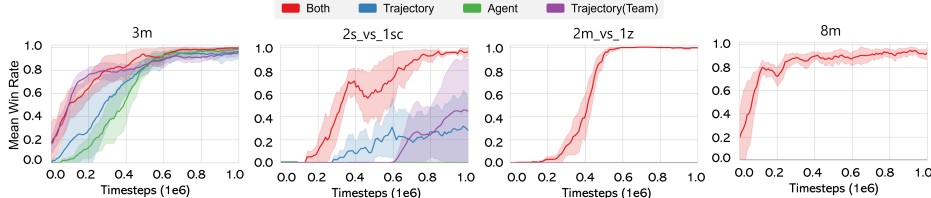

Figure 5: Comparing the performance of DPM based on preference types. It is evident that employing dual preferences yields significantly superior performance compared to using a single type of preference. **Both** refers to the case where both types of feedback are used. **Trajectory** means the case where only trajectory comparison feedback type is used. **Agent** represents the case where feedback based on comparing the actions of agents is used. **Trajectory(Team)** refers to the case where the team reward functions are trained using trajectory comparison feedback.

### 4.3 PERFORMANCE ANALYSIS OF DUAL TYPE PREFERENCES

DPM optimizes the reward model using two types of preference feedback. In this subsection, we compare the performance differences between using dual type preferences and a single preference type, to highlight the advantages of dual type feedback. Additionally, we compare the results with those obtained by training the team reward functions using trajectory feedback. Figure 5 illustrates the performances of models using only one type preference versus those incorporating both trajectory and agent comparison preferences.

The experiments are conducted on EASY scenarios in SMAC: 3m, 2m_vs_1z, 2s_vs_1sc, and 8m. For both the single and dual preference types, the total number of feedback used is the same. When using only trajectory comparison feedback, some scenarios did not converge to a high win rate or failed to solve the problem entirely. In contrast, using both types of preferences feedback(red line) leads to a convergence to an almost 100% win rate in all scenarios. Specifically, in the 2m_vs_1z and 8m scenario, the single type feedback approach showed no performance improvement at all. Furthermore, when using only agent comparison preference feedback and when training team reward functions, the performance is significantly lower, compared to DPM(both).

In online learning, unlike offline RL, the policy is trained from scratch, making initial policy training crucial. If the reward model is not well-optimized from the beginning, it is difficult to achieve good performance. Using only one type of preference often causes the reward function to fall into a local optimum. Consequently, the policy fails to learn effectively, and the quality of transitions collected subsequently is poor, making it increasingly difficult to acquire appropriate preferences in the next iteration. Therefore, using a single type of preference limits both policy and reward model training.

**Case study** : To verify the efficacy of dual type preferences in optimizing the individual reward function, we conduct a case study on a single episode of the 8m scenario. The top of Figure 6 depicts six selected scenes within the episode, describing the states and actions at these steps. The bar graphs display normalized individual reward values, scaled between 0 and 1, generated by reward models trained using different preference types. The graphs from top to bottom represent the results for training the reward models using dual preference feedback(Both), using only trajectory comparison feedback(Trajectory), and using only feedback that compares agent actions(Agent).

When comparing DPM(Both) and DPM(Trajectory), we observe that the reward models, which are trained on trajectory comparison feedback only, assign high rewards even when allied agents *die* or do useless actions such as *stop* like red bars in the graph. Similarly, when only agent comparison preference feedback is used, the individual reward model does not optimize well. For example, at step 5, agents 3 and 4 perform less cooperative actions yet receive high rewards, or at step 9, agent 1 is not participating in the battle but receives the highest reward, indicating inaccurate reward allocation.

In other words, the graphs prove that dual type preferences effectively mitigate the drawbacks of using a single preference type by leveraging the advantages of both preference types. Therefore, employing dual-preferences positively impacts the optimization of the reward model.

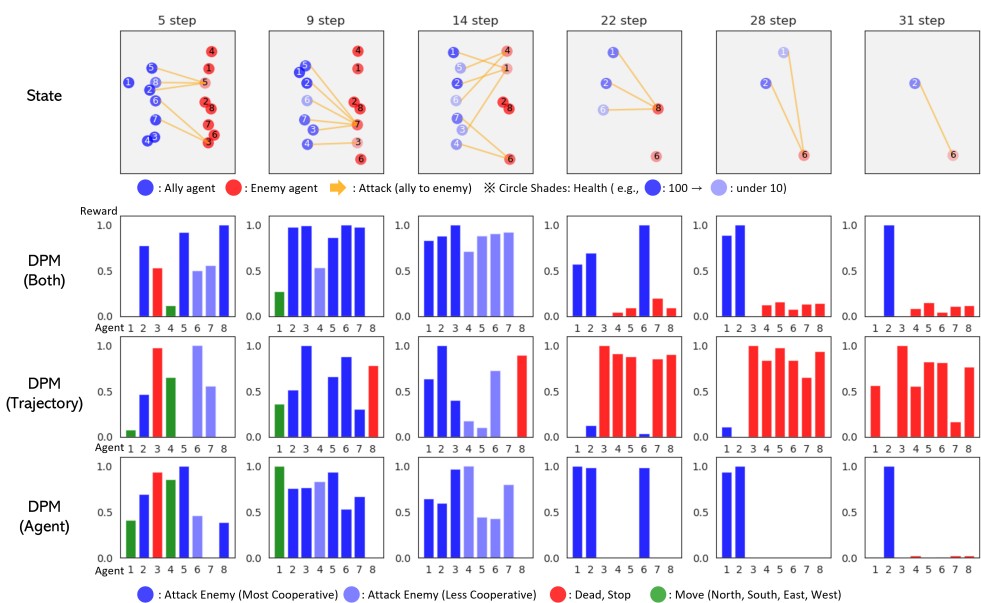

Figure 6: A case study for comparing the performance of reward models based on preference types. Comparing the rewards generated by the reward functions trained with different preference types for each of the 6 selected scenes(steps) from an episode in the `8m` scenario of SMAC.

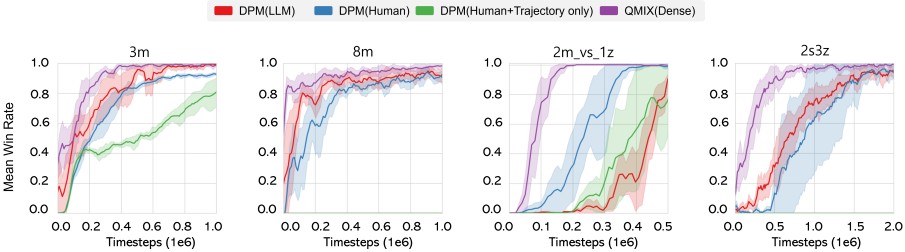

Figure 7: Performance comparison between DPM models using human preference feedback and those using an LLM preference feedback. The red line represents the DPM using the LLM feedback, the blue line represents the DPM using human preference feedback, and the green line represents the DPM trained using only one type of feedback, specifically trajectory comparison, from the human preference feedback.

## 4.4 HUMAN EXPERIMENTS

To verify whether the DPM performs well using human preference feedback instead of LLM feedback, we trained the DPM with human preference feedback in four scenarios in SMAC. The model's performance is shown in Figure 7. When compared to the case of using LLM feedback, the use of human preference feedback shows similar or even better performance. Additionally, the performance difference between using two types of preferences and not using them is significant, with the performance being much better when both types of preferences are used.

## 4.5 ABLATION STUDIES

**Impact of the amount of preference feedback on performance**: In online learning, preference feedback is collected at regular intervals and used to update the reward functions. Figure 8 (a) shows the performance based on the number of feedback collection iteration(`n_iter`), indicating a trend where performance significantly improves with more feedback collection.

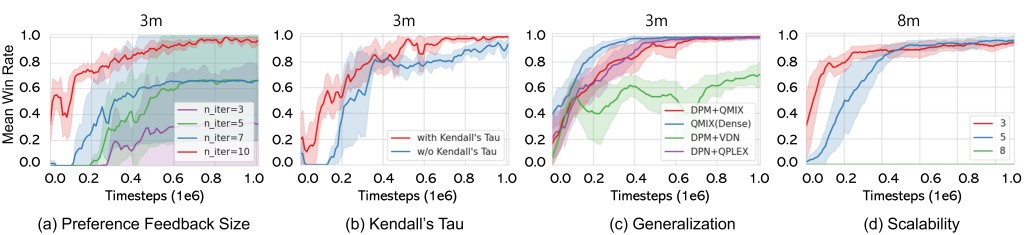

Figure 8: Ablation studies on SMAC

**Impact of the Kendall's Tau-based trajectory selection strategy**: DPM uses a strategy that employs Kendall's Tau to select the trajectory or step with the least consensus among the rankings of rewards generated by the reward functions. Figure 8 (b) compares the performance of using Kendall's Tau versus randomly selecting trajectories(or steps) for selection. The strategy used in DPM is more sample efficient and converges to a higher win rate faster compared to not using the strategy.

**The general applicability of DPM**: DPM can be applied to various MARL algorithms. Figure 8 (c) shows the results of applying DPM to dense-based MARL algorithms, QPLEX and VDN. `DPM+QPLEX` exhibits performance similar to `DPM+QMIX`, while `DPM+VDN`, although performing worse than other algorithms, shows significant performance improvement compared to when DPM is not applied.

**The scalability of DPM**: DPM obtains feedback from an LLM, requiring states and actions to be represented as prompts. However, in environments with many agents, representing all agents in the prompt can be limiting, restricting its application. To address this, a technique is employed to obtain preference feedback by utilizing only a subset of agents from the entire set. Figure 8 (d) compares the win rates of a model that collects feedback using only a subset of agents (3, 5) versus a model that collects feedback using the information from all agents in the 8m scenario. It shows a trend that using feedback from only a subset of agents, especially with fewer agents, results in higher win rates compared to using feedback from all agents. This indicates that effective training is possible by using only a subset of agents, even in environments with many agents, demonstrating that DPM is scalable.

## 5  CONCLUSION AND LIMITATION

We propose a novel approach called Dual Preference-based Multi-Agent Reinforcement Learning(DPM) for applying preference-based learning to multi-agent reinforcement learning. DPM leverages a reward model trained on preferences to transform sparse reward environments into ones akin to dense reward settings, thus addressing the sparse reward problem. Moreover, it addresses issues inherent in traditional human-based preference methods by utilizing a large language model to obtain preferences instead of relying solely on human input.

DPM differs from conventional models that solely utilize trajectory comparison feedback by introducing a preference type that compares agents' contributions through ranking. This addition helps to better optimize the reward models. We evaluate DPM in SMAC and SMACv2, which are prominent environments in multi-agent reinforcement learning. DPM demonstrates significant performance improvement compared to baselines in sparse reward settings, and its performance is comparable to that in dense reward settings. This confirms that DPM effectively addresses the sparse reward problem in MARL.

However, there exists a constraint to generate prompts due to the utilization of an LLM. It is limited to encapsulate information such as state, observation, actions, etc., within the prompt. To convert vector or image data into text form, additional prepossessing is required. Therefore, we aim for DPM to be more generally applicable across various environments through future research, such as exploring the utilization of Vision-Language Models (VLMs) to effectively substitute non-vector data such as image format data into prompts. This includes investigating methods to seamlessly incorporate data in forms other than vectors into prompts, thereby enhancing the generality of DPM.

REPRODUCIBILITY STATEMENT

For the details of environments and hyperparameters, please refer Section 4 and Appendix A. To run our method, please download the supplementary material and follow the instructions in README files. We employed pymarl2 (Hu et al., 2021) or the official codes from the authors for baselines.

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

## A  EXPERIMENTAL DETAILS

In this section, we introduce the environments used in the experiments, the baseline algorithms, as well as the hyperparameters and computational resources. Experiments are carried out on NVIDIA A6000 and GTX3090 GPUs and AMD EPYC 7313 CPU.

### A.1  ENVIRONMENTS

We conduct experiments in two environments, namely StarCraft Multi-Agent Challenge (SMAC) (Samvelyan et al., 2019) and SMACv2 (Ellis et al., 2024). All algorithms are implemented based on the open-source framework *pymarl2* (Hu et al., 2021).

We conduct experiments in the following environment:

- StarCraft Multi-Agent Challenge (SMAC) (Samvelyan et al., 2019) from `https://github.com/oxwhirl/smac` which is licensed under MIT license.
- SMACv2 (Ellis et al., 2024) from `https://github.com/oxwhirl/smacv2` which is licensed under MIT license.

All algorithms are implemented based on the open-source framework *pymarl2* (Hu et al., 2021) from `https://github.com/hijkzzz/pymarl2` which is an augmented version of *pymarl* from `https://github.com/oxwhirl/pymarl`. Both are licensed under Apache License 2.0.

#### A.1.1  SMAC

The StarCraft Multi-Agent Challenge (SMAC) is one of the benchmarks widely utilized in research to evaluate MARL algorithms. Units from the strategy video game StarCraft II engage in confrontations with each other in diver scenarios. The objective is for multiple agents to collaborate in defeating the enemies. There are multiple scenarios, each categorized into difficulty levels such as `EASY`, `HARD`, and `SuperHARD`. We primarily conduct experiments in `EASY`, and `HARD` scenarios. Table 2 provides a detailed description of the scenarios we used in our experiments.

Table 2: A detailed description of the SMAC scenario used in the experiment

| Scenario | Difficulty | Ally Units | Enemy Units | Type |
|---|---|---|---|---|
| `2s_vs_1sc` | EASY | 2 Stalkers | 1 Spine Crawler | micro-trick: alternating fire |
| `3s_vs_3z` | EASY | 3 Stalkers | 3 Zealots | micro-trick: kiting |
| `3m` | EASY | 3 Marines | 3 Marines | homogeneous & symmetric |
| `8m` | EASY | 8 Marines | 8 Marines | homogeneous & symmetric |
| `2s3z` | EASY | 2 Stalkers & 3 Zealots | 2 Stalkers & 3 Zealots | homogeneous & symmetric |
| `2m_vs_1z` | EASY | 2 Marines | 1 Zealot | micro-trick: alternating fire |
| `5m_vs_6m` | HARD | 5 Marines | 6 Marines | homogeneous & symmetric |
| `3s_vs_5z` | HARD | 3 Stalkers | 5 Zealots | micro-trick: kiting |

#### A.1.2  SMACv2

SMACv2 is proposed to address the shortcomings of SMAC, particularly in terms of its lack of stochasticity and partial observable characteristics (Ellis et al., 2024). Therefore, SMACv2 differs from SMAC in three main aspects.

First, the unit composition is randomly determined. In SMAC, the generated units are fixed, whereas in SMACv2, different types of units are randomly generated based on probabilities. The second difference lies in the observation probability of agents. In SMAC, when one agent observes an enemy, other agents can also observe the same enemy simultaneously. In contrast, in SMACv2, if one agent observes an enemy first, other agents within their observation range may not identify the same enemy,

Table 3: Calculation of the number of tokens used for DPM training and the corresponding costs

| Content | | 3m(3agents) | 2s3z(5agents) |
|---|---|---|---|
| Input tokens (per query) | Trajectory | 780~800 | 980~1,010 |
| | Agent | 1,200~1,250 | 1,800~1,850 |
| Output tokens (per query) | Trajectory | 80~100 | 80~100 |
| | Agent | 150~200 | 250~300 |
| Tokens per iteration (75 queries) | Input | 148,500~153,750 | 208,500~214,500 |
| | Output | 17,250~22,500 | 24,750~30,000 |
| Total tokens (10 iterations) | Input | 1.49M~1.54M | 2.09M~2.15M |
| | Output | 0.17M~0.26M | 0.25M~0.30M |
| Price per 1M tokens (24.11.19.) | gpt-4o | $ 2.5 (input) $ 10.00 (output) | |
| | gpt-35-turbo | $ 0.5 (input) $ 1.5 (ouptut) | |
| Cost(GPT-3.5) | Trajectory | $0.74~$0.77 | $1.04~$1.07 |
| | Agent | $0.09~$0.11 | $0.12~$0.15 |
| | Total | $0.83~$0.88 | $1.17~$1.22 |
| Cost(GPT-4) | Trajectory | $3.71~$3.84 | $5.21~$5.36 |
| | Agent | $1.72~$2.25 | $2.48~$3.00 |
| | Total | $5.44~$6.09 | $7.69~$8.36 |

even if it is present. The last distinction involves adding randomness to the location where units are spawned.

In our experiments, we keep all other factors consistent with the default settings of SMACv2, but we fix the type of agent to a single kind. For Terran, we fix the unit type to marine; for Protoss, to stalker, and for Zerg, to hydralisk.

## A.2 BASELINES

- VDN (Sunehag et al., 2017) : VDN is a value-based method and introduces a method to decompose joint value functions into individual agent value functions, addressing challenges in cooperative multi-agent reinforcement learning with a single team reward signal.

- QMIX (Rashid et al., 2018) : QMIX, one of the most widely used methods in MARL, is an extension of VDN with a mixing network for monotonic value function factorization, allowing decentralized policies to be trained centrally for better coordination and efficiency.

- QPLEX (Wang et al., 2020) : QPLEX is also a value-based method and takes a duplex dueling network for learning environment randomness and cooperative randomness independently.

- MASER (Jeon et al., 2022) : MASER is a method proposed to address sparse reward problems by generating sub-goals for agents from an experience replay buffer and providing intrinsic rewards based on these sub-goals to effectively solve the sparse reward challenge. MASER utilizes the intrinsic rewards to train both individual Q-functions and the global Q-function.

- FOX (Jo et al., 2024) : FOX is another approach to addressing the spare reward problem by proposing a formation-aware exploration framework, which provides intrinsic rewards to guide agents toward forming diverse formations, thereby reducing the exploration space and increasing efficiency.

- ICES (Li et al., 2024) : ICES introduces individual contributions as intrinsic exploration scaffolds, utilizing Bayesian surprise and privileged global information during training to address sparse reward challenges and improve cooperative exploration.

- LAIES (Liu et al., 2023) : LAIES distinguishes agent states into internal states, directly related to the agent, and external states, associated with external information, defining lazy agents based on their interaction with the environment. LAIES generates intrinsic rewards to prevent lazy agent behaviors, encouraging agents to actively interact with the environment and improving overall performance.

### A.3 LLM USAGE

The usage and costs associated with LLM utilization are summarized in Table 3. To calculate the usage, two scenarios, `3m` and `2s3z`, are compared. Each query generates approximately 1,000 tokens, resulting in an output of 100∼300 tokens. Based on this, 150 instances of preference feedback are collected per round (75 instances each), and this process is repeated a total of 10 times. Finally, the 3m scenario generates approximately 1.5M tokens, while the `2s3z` scenario produces around 2.5M tokens. Based on the pricing of GPT-3.5 and GPT-4 as of 24th November 2024, the cost is estimated to be about $1 for GPT-3.5 and $5-$8 for GPT-4.

### A.4 STRUCTURE OF REWARD FUNCTION AND TRAINING DETAILS

The reward functions adopt a structure based on linear layers, with specific architecture detailed in Table 4. In the experiments, the size of the hidden layer used is 16.

Table 4: Structure of DPM's reward functions

| Name | Type | In features | Out features |
|---|---|---|---|
| input_state | Linear | state size | hidden size |
| input_next_state | Linear | state size | hidden size |
| input_obs | Linear | observation size | hidden size |
| input_actions | Linear | action size | hidden size |
| hidden layer | Linear | hidden size × 4 | hidden size |
| output | Linear | hidden size | 1 |

The reward model consists of multiple models ($n$ models) with the same structure as described in Table4. In the experiment, three reward functions($n = 3$) are used, and the reward is calculated as the average output of these reward functions. The training of the reward functions is performed each time preference feedback is collected, totaling 10 iterations in the experiment. Once preference feedback is gathered and stored in the replay buffer, training is conducted using Equation (8) as the loss function. The reward functions use the Adam optimizer (Diederik, 2014) for optimization. After the reward model is trained, the rewards stored in the replay buffer are updated and utilized for training the Q-function.

## B ADDITIONAL EXPERIMENTS

### B.1 COMPARISON OF GROUND TRUTH REWARD AND ESTIMATED REWARD

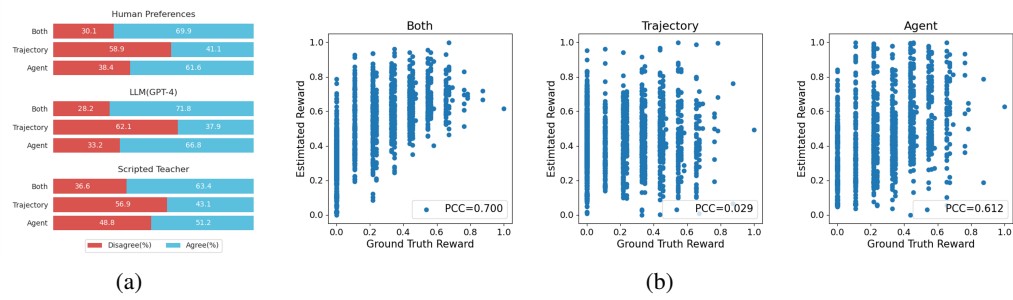

(a)                                   (b)

Figure 9: (a) A case study for comparing the performance of reward models based on preference types. (b) Pearson correlation coefficient between the ground truth reward and the estimated reward

**Preference Alignment**: To verify how well the trained reward models align with the feedback, we use human feedback, LLM-based feedback, and scripted teacher-based feedback to assess alignment. When the preferences for trajectories are calculated based on the sum of the rewards generated by the reward functions, the agreement with the actual preference feedback is shown in Figure 9(a). For comparison, 100 trajectory pairs that are not used during the training of the reward model are generated as a test set, and preference feedback is collected from humans, LLMs, and a scripted

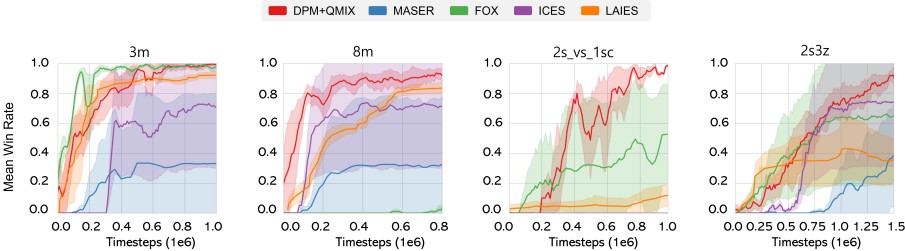

Figure 10: Experiments in a weak sparse reward setting(SMAC)

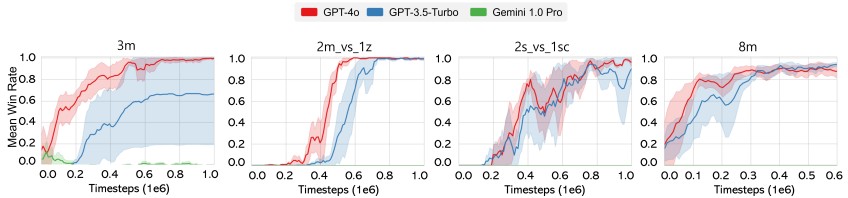

Figure 11: Comparison of DPM performance based on different types of LLMs

teacher. The trained reward model generates rewards for each trajectory, assuming that trajectories with higher rewards are more preferred. The similarity between these preferences and the actual feedback from humans, LLMs, and the scripted teacher is then evaluated. In all three feedback types, the highest agreement is observed when using two types of feedback(Both), while the lowest agreement is found when using only trajectory comparison feedback. This indicates that DPM can optimize the reward functions to align more closely with the actual feedback.

**Pearson correlation coefficient between the ground truth reward and the estimated reward**: The Pearson correlation coefficient is calculated to evaluate how well the reward model trained with preference feedback aligns with the ground truth reward. Figure 9(b) illustrates the relationship between the estimated reward and the ground truth reward for reward functions generated under different preference types. When both types of preference feedback are used (Both), the model achieves a higher PCC compared to the other cases. This indicates that the reward model is trained in a direction similar to the reward generated by the expert, ultimately contributing to the development of policies with higher win rates.

## B.2 EXPERIMENTS IN A WEAK SPARSE REWARD SETTING

Our main experiments are conducted in the hard sparse reward setting shown in Table 1, whereas much of the research on sparse reward problems has been performed in the weak sparse reward setting. Therefore, we analyze the performance of DPM and the baselines under the weak sparse reward setting. The experimental results are presented in Figure 10. The experiments are conducted in four environments of SMAC, and it is evident that the baselines perform better in the weak setting compared to the hard setting. Nevertheless, DPM consistently achieves near-optimal results across all four scenarios, with win rates approaching 100%.

## B.3 COMPARISON EXPERIMENTS ON RESULTS BASED ON LLM TYPES AND SETTINGS

DPM relies on LLMs to generate preference feedback, making the performance of the LLM a critical factor. Specifically, the LLM must have a strong understanding of long contexts to generate appropriate feedback based on the given context. Therefore, we compare the performance of DPM across different types of LLMs. The LLMs used for comparison include GPT-3.5, GPT-4 (Achiam et al., 2023), and Gemini Pro 1.0 (Team et al., 2023). The experimental results are shown in Figure

**A Prompt Example (3m – with SMAC)**

You are a helpful and honest judge of good game playing and progress in the **StarCraft Multi-Agent Challenge** game. Always answer as helpfully as possible, while being truthful.
If you don't know the answer to a question, please don't share false information.
I'm looking to have you evaluate a scenario in the **StarCraft Multi-Agent Challenge**. Your role will be to assess how much the actions taken by multiple agents in a given situation have contributed to achieving victory.

The basic information for the evaluation is as follows.

- Scenario : 3m
- Allied Team Agent Configuration : **Three marines(Marines are long-range attack units in StarCraft 2).**
- Enemy Team Agent Configuration : **Three marines(Marines are long-range attack units in StarCraft 2).**
- Situation Description : The situation involves the allied team and the enemy team engaging in combat, where victory is achieved by defeating all the enemies.
- Objective : Defeat all enemy agents while ensuring as many allied agents as possible survive.

I will provide you with two trajectories, and you should select the better trajectory based on the outcomes of these trajectories. Regarding the trajectory, it will inform you about the initial and final states, and you should select the better case based on these two trajectories.

*System Prompt*

**A Prompt Example (3m – without SMAC)**

You are a helpful and honest judge of good game playing and progress in **the game**. Always answer as helpfully as possible, while being truthful.
If you don't know the answer to a question, please don't share false information.
I'm looking to have you evaluate a scenario. Your role will be to assess how much the actions taken by multiple agents in a given situation have contributed to achieving victory.

The basic information for the evaluation is as follows.

- Allied Team Agent Configuration : **Three long-range attack units**
- Enemy Team Agent Configuration : **Three long-range attack units**
- Situation Description : The situation involves the allied team and the enemy team engaging in combat, where victory is achieved by defeating all the enemies.
- Objective : Defeat all enemy agents while ensuring as many allied agents as possible survive.

I will provide you with two trajectories, and you should select the better trajectory based on the outcomes of these trajectories. Regarding the trajectory, it will inform you about the initial and final states, and you should select the better case based on these two trajectories.

*System Prompt*

Figure 12: An example of removing SMAC from the prompt to exclude prior knowledge

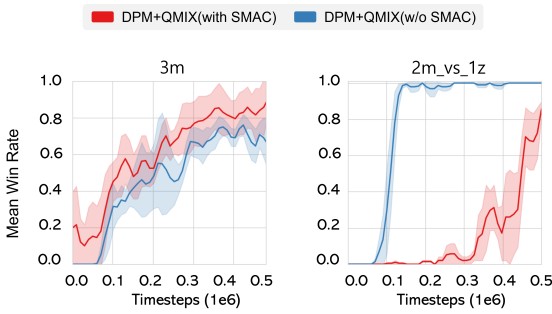

Figure 13: Comparison of DPM performance based on the presence or absence of prior knowledge in LLMs

11. Gemini Pro 1.0 failed to generate appropriate feedback in all experiments, resulting in poor DPM performance. However, while there are differences in performance between GPT-3.5 and GPT-4, both consistently generated effective preference feedback, enabling successful DPM training.

### B.4    OBSERVATION ON WHETHER THE PRIOR KNOWLEDGE OF LLM AFFECTS PERFORMANCE

Large Language Models (LLMs) are trained on vast amounts of data during pretraining, making it highly likely that they possess prior knowledge about the problems we aim to solve, including SMAC. However, for models like GPT, where training data and details are not disclosed, it is impossible to verify whether specific prior knowledge has been learned. To address this, we conducted experiments by modifying the prompts through prompt engineering to minimize prior knowledge as much as possible. Figure 12 illustrates the original prompt and the prompt with SMAC-related information

removed. The original prompt explicitly includes references to SMAC and detailed unit information. In contrast, the modified prompt omits any mention of SMAC and adjusts unit information to make it unrelated (or less directly relatable) to StarCraft.

The experimental results using the modified prompt are shown in Figure 13. The red line represents the performance of DPM with the original prompt, while the blue line represents the performance with the modified prompt. In the 3m scenario, the results were similar regardless of the presence or absence of SMAC-related information. Interestingly, in the 2m_vs_1z scenario, the absence of SMAC-related information led to even better performance. The observed difference in the 2m_vs_1z scenario suggests that the LLM may possess prior knowledge about SMAC. However, this also indicates that such prior knowledge might not always be utilized in a way that helps generate effective feedback.

## C    TRAJECTORY SELECTION STRATEGY

To obtain high-quality preference data, it is crucial to select comparison pairs appropriately. In prior PbRL research (Lee et al., 2021b), ensemble-based sampling techniques are employed. This involves assuming rewards generated by multiple reward models as preferences and selecting pairs of trajectories where the preferences do not align.

On the other hand, in DPM, individual reward functions are utilized, necessitating optimization based on individual rewards rather than global rewards which are the sum of individual rewards. However, if trajectory-based sampling, similar to single-RL, is employed, the global reward becomes the criterion, making it challenging to select appropriate trajectories. To address this issue, DPM employs Kendall's Tau (Kendall, 1938) to calculate the degree of consensus among ranking data generated from individual reward functions. The ranking is determined based on the rewards generated by the reward functions, with higher-ranked agents having higher rewards.

Since Kendall's Tau calculates the concordance between pairs of ranking data, to assess the consensus among multiple reward functions, pairwise combinations is performed, followed by averaging the results. If the value is lower than the threshold, the trajectory is added to the list for comparison. Otherwise, the trajectory is excluded from the comparison. The threshold varies with each iteration, decreasing as the iterations progress.

## D    PREFERENCE FEEDBACK COLLECTION SETTING AND RESULTS

### D.1    LLM FEEDBACK COLLECTION SETTING

DPM fundamentally uses preference feedback generated by LLMs. In online learning, the timing of feedback acquisition and the amount of feedback are crucial settings. By default, DPM is configured to repeatedly collect feedback every 1,500 episodes, gathering 150 pieces of feedback at a time. When using both types of feedback, 75 pieces are collected for each type. If only one type of feedback is used, 150 pieces are collected at a time. This process is repeated a total of 10 times, resulting in 1500 pieces of feedback used to train the reward models.

### D.2    HUMAN FEEDBACK COLLECTION SETTING AND RESULTS

All human feedback was collected by the authors, who are domain experts familiar with the MARL task and the environments. Similar to the LLM feedback collection, feedback is collected in 10 iterations, with total 150 pieces of feedback gathered in each iteration.

In the process of obtaining human feedback, preferences are determined based on the following criteria. First, in trajectory comparisons, preferences are determined by assessing scenarios where allied units retain more health while enemy units sustain significant damage. When comparing the actions of agents, the following criteria are applied. First, agents demonstrating cooperative behavior are given higher rankings (e.g., agents engaged in focused attacks are prioritized). Second, agents with high health that do not participate in engagements are assigned lower rankings. Third, agents for whom ranking is difficult are left unrated. Preferences are collected based on these criteria.

# E  PROMPT EXAMPLES

We list and discuss the prompts we employ in conducting the experiments. The prompt consists of four stages : LLM system configuration, environment description, providing information about comparisons, and task instructions.

## E.1  LLM SYSTEM CONFIGURATION

In the LLM system configuration, the LLM is endowed with roles and context awareness to enable it to generate high-quality responses.

```
You are a helpful and honest judge of good game playing and progress
in the StarCraft Multi-Agent Challenge game. Always answer as helpfully as possible,
while being truthful. If you don't know the answer to a question, please don't share
false information.

I'm looking to have you evaluate a scenario in the StarCraft Multi-Agent Challenge.
Your role will be to assess how much the actions taken by multiple agents in a given
situation have contributed to achieving victory.
```

Figure 14: Example of a system configuration prompt for the SMAC 3m scenario.

## E.2  ENVIRONMENT DESCRIPTION

The environment description encompasses a comprehensive overview of the SMAC scenario. It includes the scenario name, composition of allies, composition of adversaries, description of the situation, objectives, and other pertinent details.

```
The basic information for the evaluation is as follows.
 - Scenario : 3m
 - Allied Team Agent Configuration : Three marines
 - Enemy Team Agent Configuration : Three marines
 - Situation Description : The situation involves the allied team and the enemy team engaging in combat,
                           where victory is achieved by defeating all the enemies.
 - Objective : Defeat all enemy agents while ensuring as many allied agents as possible
               survive.

I plan to inform you about the status and actions of the agents in a single scene and I will also show
you the subsequent scene based on the agents' actions. Then, you will need to rank the agents in order
of their contribution to victory based on their actions and status.
```

Figure 15: Example of an environment description prompt for the SMAC 3m scenario.

### E.3 PROVIDING INFORMATION ABOUT THE COMPARISONS

This part describes the comparison targets for acquiring preferences. In trajectory comparison and agents comparison, separate prompts exist, each allowing for the provision of information to the LLM by altering the details in the square brackets([]), including state, actions, and other relevant information. For the trajectory comparison case, an example prompt is provided in Figure 16, and for the agent comparison case, an example prompt is given in Figure 17.

```
[Trajectory 1]
1. Final State Information
    1) Allied Agents Health : [h_f_a_1_t_1], [h_f_a_2_t_1], [h_f_a_3_t_1]
    2) Enemy Agents Health : [h_f_e_1_t_1], [h_f_e_2_t_1], [h_f_e_3_t_1]
    3) Number of Allied Deaths : [c_f_a_t_1]
    4) Number of Enemy Deaths : [c_f_e_t_1]
    5) Total Remaining Health of Allies : [r_f_a_t_1]
    6) Total Remaining Health of Enemies : [r_f_e_t_1]
2. Total Number of Steps : [step_1]

[Trajectory 2]
1. Final State Information
    1) Allied Agents Health : [h_f_a_1_t_2], [h_f_a_2_t_2], [h_f_a_3_t_2]
    2) Enemy Agents Health : [h_f_e_1_t_2], [h_f_e_2_t_2], [h_f_e_3_t_2]
    3) Number of Allied Deaths : [c_f_a_t_2]
    4) Number of Enemy Deaths : [c_f_e_t_2]
    5) Total Remaining Health of Allies : [r_f_a_t_2]
    6) Total Remaining Health of Enemies : [r_f_e_t_2]
2. Total Number of Steps : [step_2]
```

Figure 16: Example of a description of trajectories prompt for the SMAC 3m scenario.

```
1. The Scene Information
    1) Allied Agents Information
        - Ally Agent 1's Location : ([a_1_x_1],[a_1_y_1]) / Ally Agent 1's Health : [a_1_h_1]
          * Ally Agent 1's Action : [a_1_a_1]
        - Ally Agent 2's Location : ([a_2_x_1],[a_2_y_1]) / Ally Agent 2's Health : [a_2_h_1]
          * Ally Agent 2's Action : [a_2_a_1]
        - Ally Agent 3's Location : ([a_3_x_1],[a_3_y_1]) / Ally Agent 3's Health : [a_3_h_1]
          * Ally Agent 3's Action : [a_3_a_1]
    2) Enemy Agents Information
        - Enemy Agent 1's Location : ([e_1_x_1],[e_1_y_1]) / Enemy Agent 1's Health [e_1_h_1]
        - Enemy Agent 2's Location : ([e_2_x_1],[e_2_y_1]) / Enemy Agent 2's Health [e_2_h_1]
        - Enemy Agent 3's Location : ([e_3_x_1],[e_3_y_1]) / Enemy Agent 3's Health [e_3_h_1]

2. The Next Scene Information
    1) Allied Agents Information :
        - Ally Agent 1's Location : ([a_1_x_2],[a_1_y_2]) / Ally Agent 1's Health : [a_1_h_2]
        - Ally Agent 2's Location : ([a_2_x_2],[a_2_y_2]) / Ally Agent 2's Health : [a_2_h_2]
        - Ally Agent 3's Location : ([a_3_x_2],[a_3_y_2]) / Ally Agent 3's Health : [a_3_h_2]
    2) Enemy Agents Information :
        - Enemy Agent 1's Location : ([e_1_x_2],[e_1_y_2]) / Enemy Agent 1's Health [e_1_h_2]
        - Enemy Agent 2's Location : ([e_2_x_2],[e_2_y_2]) / Enemy Agent 2's Health [e_2_h_2]
        - Enemy Agent 3's Location : ([e_3_x_2],[e_3_y_2]) / Enemy Agent 3's Health [e_3_h_2]
```

Figure 17: Example of a description of state and agent actions prompt for the SMAC 3m scenario.

## E.4    TASK INSTRUCTIONS

The task instruction part provides detailed instructions regarding the output that the LLM should generate. In the trajectory comparison case, the LLM should produce preferences, while in the agent comparison case, it should generate rankings. Therefore, they have different prompt formats to facilitate these distinct tasks.

```
Your task is to inform which one is better between [Trajectory1] and [Trajectory2] based on the information
mentioned above. For example, if [Trajectory 1] seems better, output #1, and if [Trajectory 2] seems better,
output #2. If it's difficult to judge or they seem similar, please output #0.
* Important : Generally, it is considered better when fewer allied agents are killed or injured while
              inflicting more damage on the enemy.
```

Figure 18: Example of a task instruction(trajectory comparison) prompt for the SMAC 3m scenario.

```
Your task is to rank the agents in order of their contribution to victory based on their actions
and inform me of their rankings. Rankings must be displayed for all allied agents, even if a
specific agent has made no contribution. In cases where there is absolutely no contribution,
the lowest ranking should be assigned. For example, if there are three ally agents and their
contributions to victory are greatest in the order of agent 3, 1, 2,
then you should output like below :
Rank #1 : {3}
Rank #2 : {1}
Rank #3 : {2}

Moreover, if the contributions are deemed equal, assign the same rank. For example, if agent 1
and 2 contributed equally and agent 3 contributed the most, output like below :
Rank #1 : {3}
Rank #2 : {1,2}
```

Figure 19: Example of a task instruction(agents comparison) prompt for the SMAC 3m scenario.

## E.5 FULL PROMPT

```
You are a helpful and honest judge of good game playing and progress in the StarCraft Multi-Agent Challenge game.
Always answer as helpfully as possible, while being truthful.
If you don't know the answer to a question, please don't share false information.
I'm looking to have you evaluate a scenario in the StarCraft Multi-Agent Challenge. Your role will be to assess
how much the actions taken by multiple agents in a given situation have contributed to achieving victory.

The basic information for the evaluation is as follows.

- Scenario : 3m
- Allied Team Agent Configuration : Three marines(Marines are long-range attack units in StarCraft 2).
- Enemy Team Agent Configuration : Three marines(Marines are long-range attack units in StarCraft 2).
- Situation Description : The situation involves the allied team and the enemy team engaging in combat,
                          where victory is achieved by defeating all the enemies.
- Objective : Defeat all enemy agents while ensuring as many allied agents as possible survive.

I will provide you with two trajectories, and you should select the better trajectory based on the outcomes of
these trajectories. Regarding the trajectory, it will inform you about the initial and final states,
and you should select the better case based on these two trajectories.

[Trajectory 1]
1. Final State Information
    1) Allied Agents Health : [h_f_a_1_t_1], [h_f_a_2_t_1], [h_f_a_3_t_1]
    2) Enemy Agents Health : [h_f_e_1_t_1], [h_f_e_2_t_1], [h_f_e_3_t_1]
    3) Number of Allied Deaths : [c_f_a_t_1]
    4) Number of Enemy Deaths : [c_f_e_t_1]
    5) Total Remaining Health of Allies : [r_f_a_t_1]
    6) Total Remaining Health of Enemies : [r_f_e_t_1]
2. Total Number of Steps : [step_1]

[Trajectory 2]
1. Final State Information
    1) Allied Agents Health : [h_f_a_1_t_2], [h_f_a_2_t_2], [h_f_a_3_t_2]
    2) Enemy Agents Health : [h_f_e_1_t_2], [h_f_e_2_t_2], [h_f_e_3_t_2]
    3) Number of Allied Deaths : [c_f_a_t_2]
    4) Number of Enemy Deaths : [c_f_e_t_2]
    5) Total Remaining Health of Allies : [r_f_a_t_2]
    6) Total Remaining Health of Enemies : [r_f_e_t_2]
2. Total Number of Steps : [step_2]

Your task is to inform which one is better between [Trajectory1] and [Trajectory2] based on the information
mentioned above. For example, if [Trajectory 1] seems better, output #1, and if [Trajectory 2] seems better,
output #2. If it's difficult to judge or they seem similar, please output #0.
* Important : Generally, it is considered better when fewer allied agents are killed or injured while
              inflicting more damage on the enemy.
```

Figure 20: Example of a full trajectory comparison prompt for the SMAC 3m scenario.

```
You are a helpful and honest judge of good game playing and progress
in the StarCraft Multi-Agent Challenge game. Always answer as helpfully as possible,
while being truthful.  If you don't know the answer to a question, please don't share
false information.

I'm looking to have you evaluate a scenario in the StarCraft Multi-Agent Challenge.
Your role will be to assess how much the actions taken by multiple agents in a given
situation have contributed to achieving victory.

The basic information for the evaluation is as follows.
 - Scenario : 3m
 - Allied Team Agent Configuration : Three marines
 - Enemy Team Agent Configuration : Three marines
 - Situation Description : The situation involves the allied team and the enemy team engaging in combat,
                           where victory is achieved by defeating all the enemies.
 - Objective : Defeat all enemy agents while ensuring as many allied agents as possible
                survive.

I plan to inform you about the status and actions of the agents in a single scene and I will also show
you the subsequent scene based on the agents' actions. Then, you will need to rank the agents in order
of their contribution to victory based on their actions and status.

1. The Scene Information
  1) Allied Agents Information
      - Ally Agent 1's Location : ([a_1_x_1],[a_1_y_1]) / Ally Agent 1's Health : [a_1_h_1]
        * Ally Agent 1's Action : [a_1_a_1]
      - Ally Agent 2's Location : ([a_2_x_1],[a_2_y_1]) / Ally Agent 2's Health : [a_2_h_1]
        * Ally Agent 2's Action : [a_2_a_1]
      - Ally Agent 3's Location : ([a_3_x_1],[a_3_y_1]) / Ally Agent 3's Health : [a_3_h_1]
        * Ally Agent 3's Action : [a_3_a_1]
  2) Enemy Agents Information
      - Enemy Agent 1's Location : ([e_1_x_1],[e_1_y_1]) / Enemy Agent 1's Health [e_1_h_1]
      - Enemy Agent 2's Location : ([e_2_x_1],[e_2_y_1]) / Enemy Agent 2's Health [e_2_h_1]
      - Enemy Agent 3's Location : ([e_3_x_1],[e_3_y_1]) / Enemy Agent 3's Health [e_3_h_1]

2. The Next Scene Information
  1) Allied Agents Information :
      - Ally Agent 1's Location : ([a_1_x_2],[a_1_y_2]) / Ally Agent 1's Health : [a_1_h_2]
      - Ally Agent 2's Location : ([a_2_x_2],[a_2_y_2]) / Ally Agent 2's Health : [a_2_h_2]
      - Ally Agent 3's Location : ([a_3_x_2],[a_3_y_2]) / Ally Agent 3's Health : [a_3_h_2]
  2) Enemy Agents Information :
      - Enemy Agent 1's Location : ([e_1_x_2],[e_1_y_2]) / Enemy Agent 1's Health [e_1_h_2]
      - Enemy Agent 2's Location : ([e_2_x_2],[e_2_y_2]) / Enemy Agent 2's Health [e_2_h_2]
      - Enemy Agent 3's Location : ([e_3_x_2],[e_3_y_2]) / Enemy Agent 3's Health [e_3_h_2]

Your task is to rank the agents in order of their contribution to victory based on their actions
and inform me of their rankings. Rankings must be displayed for all allied agents, even if a
specific agent has made no contribution. In cases where there is absolutely no contribution,
the lowest ranking should be assigned. For example, if there are three ally agents and their
contributions to victory are greatest in the order of agent 3, 1, 2,
then you should output like below :
Rank #1 : {3}
Rank #2 : {1}
Rank #3 : {2}

Moreover, if the contributions are deemed equal, assign the same rank. For example, if agent 1
and 2 contributed equally and agent 3 contributed the most, output like below :
Rank #1 : {3}
Rank #2 : {1,2}
```

Figure 21: Example of a full agents comparison prompt for the SMAC 3m scenario.

## E.6 EXAMPLE AND RESPONSE IN THE SMAC 3M SCENARIO

### E.6.1 TRAJECTORY COMPARISON

---

**A Prompt Example (3m – Trajectory)**

You are a helpful and honest judge of good game playing and progress in the StarCraft Multi-Agent Challenge game. Always answer as helpfully as possible, while being truthful.
If you don't know the answer to a question, please don't share false information.
I'm looking to have you evaluate a scenario in the StarCraft Multi-Agent Challenge. Your role will be to assess how much the actions taken by multiple agents in a given situation have contributed to achieving victory.

The basic information for the evaluation is as follows.

- Scenario : 3m
- Allied Team Agent Configuration : Three marines(Marines are long-range attack units in StarCraft 2).
- Enemy Team Agent Configuration : Three marines(Marines are long-range attack units in StarCraft 2).
- Situation Description : The situation involves the allied team and the enemy team engaging in combat, where victory is achieved by defeating all the enemies.
- Objective : Defeat all enemy agents while ensuring as many allied agents as possible survive.

I will provide you with two trajectories, and you should select the better trajectory based on the outcomes of these trajectories. Regarding the trajectory, it will inform you about the initial and final states, and you should select the better case based on these two trajectories.

[Trajectory 1]
1. Final State Information
   1) Allied Agents Health : 20, 40, 60
   2) Enemy Agents Health : 0, 0, 0
   3) Number of Allied Deaths : 0
   4) Number of Enemy Deaths : 3
   5) Total Remaining Health of Allies : 120
   6) Total Remaining Health of Enemies : 0

[Trajectory 2]
1. Final State Information
   1) Allied Agents Health : 40, 0, 0
   2) Enemy Agents Health : 0, 0, 0
   3) Number of Allied Deaths : 2
   4) Number of Enemy Deaths : 3
   5) Total Remaining Health of Allies : 40
   6) Total Remaining Health of Enemies : 0

Your task is to inform which one is better between [Trajectory1] and [Trajectory2] based on the information mentioned above. For example, if [Trajectory 1] seems better, output #1, and if [Trajectory 2] seems better, output #2. If it's difficult to judge or they seem similar, please output #0.
※ Important : Generally, it is considered better when fewer allied agents are killed or injured while inflicting more damage on the enemy.

Please also let me know the reason for your decision.

*System Prompt* / *Trajectory #1* / *Trajectory #2* / *Mission*

---

Figure 22: Example of a trajectory comparison prompt for the SMAC 3m scenario.

---

**Response Example (3m – Trajectory / GPT4)**

#1
**Reason**: In [Trajectory 1], all allied agents survive, and the total remaining health of allies is 120. In [Trajectory 2], two allied agents are killed, leaving only one agent with 40 health. Since [Trajectory 1] achieves the objective of defeating all enemy agents while ensuring the survival and higher health of allied agents, it is clearly the better outcome.

---

Figure 23: Example of a trajectory comparison response for the SMAC 3m scenario.

## E.6.2 ACTION COMPARISON

---

**A Prompt Example (3m – Action Comparison)**

You are a helpful and honest judge of good game playing and progress
in the StarCraft Multi-Agent Challenge game. Always answer as helpfully as possible, while being truthful. If you don't know the answer to a question, please don't share false information.

I'm looking to have you evaluate a scenario in the StarCraft Multi-Agent Challenge. Your role will be to assess how much the actions taken by multiple agents in a given situation have contributed to achieving victory.

The basic information for the evaluation is as follows.

- Scenario : 3m
- Allied Team Agent Configuration : Three marines(Marines are long-range attack units in StarCraft 2).
- Enemy Team Agent Configuration : Three marines(Marines are long-range attack units in StarCraft 2).
- Situation Description : The situation involves the allied team and the enemy team engaging in combat, where victory is achieved by defeating all the enemies.
- Objective : Defeat all enemy agents while ensuring as many allied agents as possible survive.
 * Important notice : To win, focused attacks are necessary. Therefore, prioritize agents contributing to attacking the same enemy. Also, assign lower contribution scores to agents with high health who are not engaging enemies.

I plan to inform you about the status and actions of the agents in a single scene and I will also show you the subsequent scene based on the agents' actions. Then, you will need to rank the agents in order of their contribution to victory based on their actions and status.

1. The Scene Information
   1) Allied Agents Information
      - Ally Agent 1's Location : (0.0,0.0) / Ally Agent 1's Health : 20 / * Ally Agent 1's Action : Attack Enemy #1
      - Ally Agent 2's Location : (0.2,0.0) / Ally Agent 2's Health : 40 / * Ally Agent 2's Action : Move West
      - Ally Agent 3's Location : (0.0,0.2) / Ally Agent 3's Health : 80 / * Ally Agent 3's Action : Attack Enemy #2
   2) Enemy Agents Information
      - Enemy Agent 1's Location : (0.5,0.5) / Enemy Agent 1's Health 10
      - Enemy Agent 2's Location : (0.5,0.7) / Enemy Agent 2's Health 50
      - Enemy Agent 3's Location : (0.3,0.5) / Enemy Agent 3's Health 100

2. The Next Scene Information
   1) Allied Agents Information :
      - Ally Agent 1's Location : (0.0,0.0) / Ally Agent 1's Health : 20
      - Ally Agent 2's Location : (0.2,-0.1) / Ally Agent 2's Health : 40
      - Ally Agent 3's Location : (0.0,0.2) / Ally Agent 3's Health : 60
   2) Enemy Agents Information
      - Enemy Agent 1's Location : (0.0,0.0) / Enemy Agent 1's Health 0
      - Enemy Agent 2's Location : (0.5,0.7) / Enemy Agent 2's Health 50
      - Enemy Agent 3's Location : (0.3,0.5) / Enemy Agent 3's Health 100

Your task is to rank the agents in order of their contribution to victory based on their actions and inform me of their rankings. Rankings must be displayed for all allied agents, even if a specific agent has made no contribution. In cases where there is absolutely no contribution, the lowest ranking should be assigned. For example, if there are three ally agents and their contributions to victory are greatest in the order of agent 3, 1, 2, then you should output like below :
Rank #1 : {3}
Rank #2 : {1}
Rank #3 : {2}

Moreover, if the contributions are deemed equal, assign the same rank. For example, if agent 1 and 2 contributed equally and agent 3 contributed the most, output like below :
Rank #1 : {3}
Rank #2 : {1,2}

Please also let me know the reason for your decision.

*(Side labels: System Prompt / State and Actions / Next State / Mission)*

Figure 24: Example of an action comparison prompt for the SMAC 3m scenario.

**Response Example (3m – Action Comparison / GPT4)**

**Rankings:**
Rank #1: {1}
Rank #2: {3}
Rank #3: {2}
**Reason:**
•**Ally Agent 1**: Agent 1 attacked Enemy #1, resulting in Enemy #1 being eliminated in the next scene. This is a direct and decisive contribution to victory.
•**Ally Agent 3**: Agent 3 attacked Enemy #2, but there was no observable reduction in Enemy #2's health in the next scene, possibly due to insufficient damage. While it showed engagement, the impact was less significant than Agent 1's.
•**Ally Agent 2**: Agent 2 moved west, which did not contribute to engaging the enemy or securing an advantage in combat. This action provided no direct contribution to the objective, making it the least impactful.

Figure 25: Example of an action comparison response for the SMAC 3m scenario.

