# OpenReview forum: "DPM: Dual Preferences-based Multi-Agent Reinforcement Learning"
_ICLR.cc/2025/Conference — Submitted to ICLR 2025_

### Official Review · Reviewer_NVua · 2024-11-01

**Soundness:** 3
**Presentation:** 3
**Contribution:** 3
**Rating:** 5
**Confidence:** 4

**Summary:**

This paper addresses the sparse reward problem in cooperative MARL utilizing PbRL techniques. Specifically, besides comparing trajectories like SARL settings, this paper compares / ranks agents in a state to better learn the reward models. It also uses LLMs for automated comparison to reduce the reliance on human input, and achieves good results on benchmarks SMAC and SMACv2.

**Strengths:**

- This paper is well written and easy to follow.
- The idea of extending PbRL into multi-agent settings by comparing / ranking agents is inspiring, which can be considered as an implicit credit assignment.
- Experiment results show that DPM can well “reconstruct” the dense rewards in SMAC scenarios, thus achieving comparable performance with the baseline that learns from original dense rewards.
- This paper provides an example of transforming SMAC game information into texts, which can be beneficial for relevant future research.

**Weaknesses:**

- The reward calculation for running QMIX is not clearly illustrated. According to Line 307, intrinsic rewards are $\hat{r}$ and $\hat{R}\_t$, which are not defined in previous texts. I assume that $\hat{r} = \hat{R}\_t = \sum\_{j \in N} \hat{r}_{\psi} (s_t, a_t^j, o_t^j)$ according to Line 229. However, this reward function does not take $a_t^{-j}$ as input, which does not align with MARL settings where the rewards for individual actions depend on teammates’ actions.
- Even if the individual rewards are optimal, the sum operation will lose their information. For example, the reward sum is the same when individual rewards are (1, 0, …, 0) and (0, 0, …, 1). Authors should develop techniques to handle this problem, or apply DPM to MARL algorithms, like MAPPO[1], that can directly learn from individual rewards.
- There has already been a series of work[2,3,4,5] that utilizes LLMs to build dense rewards to solve sparse reward tasks. This paper should introduce these work and include them as baselines.
- StarCraftII is a popular game and SMAC is a popular benchmark. It is likely that LLMs already know the original dense reward function and use it for comparison. However, we should be more interested in whether DPM can solve novel sparse reward MARL tasks. It would be better if the authors discuss about this open problem.
- It seems unnecessary to include both versions of SMAC as benchmark, especially when the earlier version has “deterministic limitations”.
- More experiment details should be highlighted, including baseline introduction, the process of reward model learning, results with different LLMs, and the cost of LLM api for conducting the entire research and a single DPM running.

[1] Chao Yu, et al. The Surprising Effectiveness of PPO in Cooperative, Multi-Agent Games.

[2] Tianbao Xie, Siheng Zhao, et al. Text2Reward: Reward Shaping with Language Models for Reinforcement Learning.

[3] Yecheng Jason Ma, et al. Eureka: Human-Level Reward Design via Coding Large Language Models.

[4] Shengjie Sun, Runze Liu, et al. A Large Language Model-Driven Reward Design Framework via Dynamic Feedback for Reinforcement Learning.

[5] Yun Qu, et al. Choices are More Important than Efforts: LLM Enables Efficient Multi-Agent Exploration.

**Questions:**

See Weaknesses.

---

> ### Author Response · Authors · 2024-11-20
>
> Dear reviewer, we are sincerely grateful for your comprehensive feedback and constructive comments. We also appreciate the opportunity to respond to your concerns. Our responses are presented below.
>
> ---
>
> ### **Weakness**
> **1. Regarding the reward function**
>
> Based on your feedback, we have added further explanation about the notation used in the reward formula in the main text (lines 298~299). As the reviewer mentioned, while the actions of other agents can influence the reward, conversely, there are environments where only the individual actions of agents impact the reward. In the case of SMAC, state changes resulting from the individual actions of agents are reflected in the team reward. Moreover, many researches [1],[2],[3] that use intrinsic individual rewards do not incorporate $a^{-j}$, and following this reference, we designed the reward function to calculate individual rewards considering only $a^j$, without $a^{-j}$.
>
> **2. Utilization of individual rewards**
>
> As you pointed out, information loss may occur when summing individual rewards. Therefore, research that focuses on using individual rewards to train individual agent policies is necessary, and we are considering this as a direction for future work. However, from the perspective of generalization applicable to various MARL algorithms, most MARL algorithms are based on team rewards. Thus, generating team rewards contributes to generalizability. To verify the generalization of DPM, we conducted an ablation study using QPLEX and VDN as backbone networks. This confirmed that DPM can be applied to various MARL algorithms.
>
> **3. Related works addressing the sparse reward problem using LLMs**
>
> Thank you for recommending the related studies. We have added the studies mentioned by the reviewer [4, 5, 6, 7] to the related work section (lines 108~116). However, studies [4, 5, 6] were all applied to single-agent scenarios, which poses challenges for directly applying their methods to MARL. [7] was published after we submitted the paper (October 3), so we were unable to include it in the related works section at the time. However, thanks to the reviewer's feedback, we have now added it. Moreover, we hope to include this paper as a baseline. Unfortunately, incorporating it as a baseline is limited due to the tight schedule. (The experiments are currently ongoing, and if completed, we will include the results.)
>
> **4. Regarding Prior Knowledge (LLM)**
>
> Please refer to item 3 in the Common Response.
>
> **5. The necessity of SMAC**
>
> Although SMAC has issues with being deterministic, it is still a representative environment widely used in MARL research. The availability of multiple scenarios allows for the evaluation of performance across a variety of situations (e.g., heterogeneous, cooperative, scalability). Therefore, we consider it necessary. Additionally, SMACv2 addresses some of the limitations of SMAC, which justifies using both environments.
>
> **6. Experiment details**
>
> We have added more detailed explanations about the experiments in the main text. Please refer to common response 1 for further details.
>
> ---
>
> ### **References**
> [1] Ma, Zixian, et al. "Elign: Expectation alignment as a multi-agent intrinsic reward." Advances in Neural Information Processing Systems 35 (2022): 8304-8317.
>
> [2] Du, Yali, et al. "Liir: Learning individual intrinsic reward in multi-agent reinforcement learning." Advances in Neural Information Processing Systems 32 (2019).
>
> [3] Jing, Xiao, et al. "Divide and Explore: Multi-Agent Separate Exploration with Shared Intrinsic Motivations." (2021).
>
> [4] Tianbao Xie, Siheng Zhao, et al. Text2Reward: Reward Shaping with Language Models for Reinforcement Learning.
>
> [5] Yecheng Jason Ma, et al. Eureka: Human-Level Reward Design via Coding Large Language Models.
>
> [6] Shengjie Sun, Runze Liu, et al. A Large Language Model-Driven Reward Design Framework via Dynamic Feedback for Reinforcement Learning.
>
> [7] Yun Qu, et al. Choices are More Important than Efforts: LLM Enables Efficient Multi-Agent Exploration.
>
> ---
>
> We hope our responses have addressed your concerns. If there are any remaining issues, we would be grateful and pleased to discuss them during the discussion period. Thank you once again for your time and invaluable feedback.

---

> > ### Comment · Reviewer_NVua · 2024-11-26
> >
> > Thanks for the detailed reply. I have no further questions and will maintain my rating.

---

> ### Author Response · Authors · 2024-11-28
>
> We sincerely thank you for taking the time to review our paper.
>
> If you have any remaining concerns, we would be grateful if you could share them with us.
>
> Best Regards,
>
> The Authors.

---

### Official Review · Reviewer_YJrV · 2024-11-02

**Soundness:** 2
**Presentation:** 4
**Contribution:** 2
**Rating:** 6
**Confidence:** 3

**Summary:**

This paper proposes a preferences-based MARL algrorithm named DPM which aims at solving the sparse reward problems in MARL by utlising LLM to label trajectory preferences across trajectories and agent preferences within one time step. The preferences across agents help to identify agents' contribution within one step. The experimental results on SMAC and SMACv2 show that DPM outperform many strong baselines in sparse reward setting and achieve close performance to QMIX in dense reward setting.

**Strengths:**

The paper is well written and most parts of the paper are easy to follow. The paper conducts a comprehensive set of experiments with ablation studies. The idea to access an agent's contribution to a team by utlising agents preference labelled by LLM is novel and the results demonstrate its effectiveness.

**Weaknesses:**

Some essential details or results are missing, so I scored the paper's soundness as fair:

1. How long is a trajectory used in trajectory comparison. Is it a complete episode or part of the episode? If it is the former, how does it align with the extrinsic reward provided by the environment in sparse reward setting.

2.  Could you compare the DPM performance based on human feedback with LLM feedback on the same graph?

3. In line 344, you mention spaese-reward based algorithms perform poorly becasue of a harsher sparse setting, could you compare the sparse settings explicitly?

4. Could you analyse the reason that DPM outperforms QMIX dense in 10gen-terran? does it mean LLM can do better than extrinsic dense rewards?

5. Could you provide some actual examples of the LLM's prompt and cooresponding response on trajectories and agents comparison?

6. Do you think the results of Figure 8(d) contradicts with your motivation that we should assess the agent's individual contribution to the team for better performance? How can the trend that "feedback from only a subset of agents, especially with fewer agents, results in higher win rates compared to using feedback from all agents." support your motivation. Could you analyse the reason for the trend? Also, could you suggest a way to selecet the number of agents we should compare in large scale games and which agents should be compared?

Some potential limitations are not discussed or addressed, so I scored the paper's contribution as fair:
7. How do you think the tradeoff between the cost (time and money) and the performance achieved by DPM? Given the same number of floating-point arithmetic calculations, will there be alternative apporach?
8. Does DPM rely on the prior knowledge of the games? In many sparse reward problems, even human experts may find it diffcult to compare agents' contribution within one step.

I am willing to alter my socres if my questions are well adressed.

**Questions:**

Please refer to my weaknesses part.

---

> ### Author Response · Authors · 2024-11-20
>
> Dear reviewer, we sincerely appreciate your careful review and valuable comments. We are glad to have the opportunity to address your feedback. We present our responses below.
>
> ---
> ### **Weakness**
>
> **1. Length setting of trajectories for feedback**
>
> The latter of the two methods mentioned by the reviewer applies here, where the length changes dynamically during each trajectory comparison, and pairs are formed as part of the episode.
>
> **2. Performance comparison on human feedback**
>
> Please refer to item 2 in the Common Response.
>
> **3. Explanation of the sparse reward settings (weak and hard)**
>
> The explanation for the sparse-reward setting is provided in Table 1. In the weak sparse case, rewards are given even when allies or enemy agents die, whereas in the hard sparse case, no rewards are given for agent deaths, and rewards are only provided at the end of the episode (i.e., when the outcome of the battle is determined). Additionally, we have added experiments in the weak sparse environment to assist with your evaluation, as shown in Figure 10 (lines 901-909). Please refer to the common response item 6 for more details.
>
> **4. Performance analysis in the `10gen_terran` scenario**
>
> The reward (extrinsic) provided by the environment is also designed by human experts, which means it cannot guarantee optimality. In the case of `10gen-terran`, it can be observed that the reward function generated by DPM is more suitably optimized for the task. There are occasional instances where the reward generated by the PbRL reward function outperforms that learned solely based on the environment-provided rewards. Examples such as [1-3] are provided.
>
> **5. Examples of LLM prompt and the response**
>
> Examples of LLM prompts and responses have been added in Appendix E.6 (Figure 22~25). The LLM prompt is divided into three parts: the system part, which explains the environment; the part mentioning the trajectory to compare (or the agent's actions); and the part defining what the LLM is supposed to do. An example of the output is provided in Figure 23 and Figure 25, demonstrating that the LLM generates an output consistent with the given instructions.
>
> **6. About Figure 8(d)**
>
> Ranking the actions of individual agents does not require comparing all agents at each step. Even if we only compare specific agents' actions, the reward function can still be learned in a pair-wise manner using the rankings of the actions, meaning there are no significant limitations to comparing only a subset of agents. The key point is distinguishing valuable actions among the agents' actions, which can be achieved using only a subset of the actions rather than the entire set. On the contrary, increasing the number of actions to rank can actually reduce the accuracy of the LLM and human feedback. Therefore, it is not necessary to compare every action—having preferences between actions is sufficient to learn the reward function. In other words, what is required to train the reward model is the relative value of actions. Even if we do not have preferences for every agent's action, the preferences for some actions are enough to model the reward function. We selected agents randomly, and the number of agents for comparison was manually set. To determine the appropriate number of agents on a larger scale, it is necessary to compare LLM rankings with human rankings and adjust the number accordingly.
>
> **7. Trade-off between cost (time and money) and performance**
>
> Generally, PbRL performs better as the number of preference feedback increases, improving both convergence speed and overall performance. However, the number of feedbacks is directly proportional to the cost (for both human and LLM). DPM can be considered cost-efficient compared to obtaining preferences solely from trajectory pairs, as it delivers better performance with the same number of preference feedback. Moreover, using an LLM to collect preference feedback is more cost-effective than obtaining feedback from humans. Since DPM demonstrates performance similar to or even better than human feedback (Figure 7, lines 456-470) despite its lower cost, it can be considered cost-efficient. (The scripted teacher method, which calculates preferences using dense rewards, is excluded as it is limited in practical application.)
>
> **8. Dependency on prior knowledge**
>
> Please refer to item 3 in the Common Response.
>
> ---
> ### **References**
>
> [1] Zhang, Zhilong, et al. Flow to better: Offline preference-based reinforcement learning via preferred trajectory generation. ICLR 2023.
>
> [2] Kim, Changyeon, et al. Preference transformer: Modeling human preferences using transformers for rl. ICLR 2023.
>
> [3] Zhu, Mingye, et al. LIRE: listwise reward enhancement for preference alignment. arXiv preprint arXiv:2405.13516 (2024).
>
> ---
> We hope our responses have addressed your concerns. If there are any remaining issues, we would be thankful if you could share them. Thank you once again for your time and effort.

---

> > ### Comment · Reviewer_YJrV · 2024-11-28
> >
> > Thanks for the clarification. I am inclined towards accepting this paper and I have updated the score to indicate the same

---

> ### Author Response · Authors · 2024-11-28
>
> We sincerely appreciate your support and time you have dedicated to evaluating our paper.
>
> If you have any further unresolved issues, we would greatly appreciate it if you could share them with us.
>
> Best Regards,
>
> The Authors.

---

### Official Review · Reviewer_an6t · 2024-11-03

**Soundness:** 2
**Presentation:** 3
**Contribution:** 2
**Rating:** 5
**Confidence:** 4

**Summary:**

This paper addresses the challenge of sparse rewards in multi-agent reinforcement learning (MARL) by introducing Dual Preferences-based Multi-Agent Reinforcement Learning (DPM). DPM is a dual preferences-based method that evaluates both complete trajectories and individual agent contributions. Experimental results on the StarCraft Multi-Agent Challenge (SMAC) and SMACv2 demonstrate that DPM significantly outperforms existing sparse-reward baselines.

**Strengths:**

1. The preference-based method appears to be novel in MARL settings and it is shown effective in the experiments.
2. The case study and ablation study section provides comprehensive insights.

**Weaknesses:**

1. My primary concern lies in evaluation.

    - DPM is mainly evaluated against exploration methods in MARL. This comparison lacks fairness since DPM receives additional environment knowledge through LLM prompt templates (including game objectives, cues on how to win the game, and state interpretation) while baseline algorithms (except densely rewarded QMIX) do not have access to such information.
    - The evaluation relies solely on StarCraft-based benchmarks, lacking diversity. Additional benchmarks with different characteristics would strengthen the evaluation.

2. The motivation for agent ranking requires better justification. The direct comparison of agent actions in partially observable or heterogeneous settings (where agents have different types and abilities) needs more explanation.

3. The use of multiple reward functions introduces additional training overhead. Additionally, since preference pairs are generated from the replay buffer throughout the training, the computational costs of LLM queries should be reported.

4. Minor comments:
    - Line 179, typo with actions $o_t$

**Questions:**

1. In the manuscripts, the author mentioned DPM adopts multiple reward models. How many distinct reward models does DPM employ? What are their specific roles (trajectory vs. agent preferences)? And how do these models differ in structure or function?
2. What is the required number of preference pairs per scenario? And how does the number of pairs affect performance?

---

> ### Author Response · Authors · 2024-11-20
>
> Dear reviewer, thank you for your careful review and constructive comments. We greatly appreciate the opportunity to address your feedback. We present our responses below.
>
> ---
>
> ### **Weakness**
>
> **1. Experimental design (fairness)**
>
> Please refer to item 2 and 3 in the Common Response.
>
> **2. The motivation for agent ranking**
>
> In MARL, since there are multiple agents, the actions of the agents in a single scene can be either meaningful or insignificant. However, traditional methods that rely solely on trajectory make it difficult to assess the quality of individual agents' actions. To address this, we utilized ranking to relatively evaluate the value of agents' actions. This approach enables a more detailed analysis of the value of their actions, leading to better optimization of the reward function, as demonstrated in our experiments. The method of ranking agents' actions is similar to the action preference approach proposed in [1]. Action preference involves comparing two actions in a specific state to determine which is preferred. This approach helps mitigate the temporal credit assignment problem inherent in trajectory preference, which involves identifying which states or actions contribute to the overall performance. However, in a single-agent environment, it is challenging to obtain transitions with different actions for the same state. In contrast, in a multi-agent environment, multiple actions coexist in a single state, making it possible to incorporate action preference into trajectory preference, which contributes to performance improvement.
>
> **3. About multiple reward functions (with Question #1)**
>
> We use multiple reward functions, but contrary to what the reviewer mentioned, these reward functions do not serve different roles. They have the same role and identical structures. By training these reward functions independently and using the average of their outputs, we aim to mitigate potential biases that could arise from having only one reward model. In this experiment, we used three identical reward models. This approach is common in PbRL and has been utilized as an ensemble learning technique in previous studies such as PEBBLE[2], RUNE[3], and RIME[4]. This ensemble method enhances the robustness of reward generation. Additionally, the time required for reward training is significantly less than that for policy training, so the computational overhead for reward modeling is minimal.
>
> **4. Typo correction**
>
> * Thank you for pointing out the typos. I have corrected the mentioned typos as well as other typos throughout the paper.
>
> ---
>
> ### **Questions**
>
> **1. Amount of preference feedback**
>
> The specific preference pairs used for training are detailed in the Appendix D (lines 1005–1025). In each feedback collection iteration, we gathered 150 feedback items (75 trajectory feedback and 75 action feedback), and through a total of 10 iterations, we used 1,500 preference pairs for training. Additionally, the impact of the number of preference feedback items on performance is analyzed in Ablation 1(Figure 8(a) and lines 483-485). Here, we examine the performance differences as the number of feedback collection iterations (i.e., the amount of feedback) changes. The experimental results show that the more feedback collected (the more iterations), the better the performance.
>
> ---
>
> ### **References**
>
> [1] Wirth, Christian, et al. "A survey of preference-based reinforcement learning methods." Journal of Machine Learning Research 18.136 (2017): 1-46.
>
> [2] Lee, Kimin, Laura Smith, and Pieter Abbeel. "Pebble: Feedback-efficient interactive reinforcement learning via relabeling experience and unsupervised pre-training." arXiv preprint arXiv:2106.05091 (2021).
>
> [3] Liang, Xinran, et al. "Reward uncertainty for exploration in preference-based reinforcement learning." arXiv preprint arXiv:2205.12401 (2022).
>
> [4] Cheng, Jie, et al. "RIME: Robust Preference-based Reinforcement Learning with Noisy Preferences." arXiv preprint arXiv:2402.17257 (2024).
>
> ---
> We hope our responses have addressed your concerns. If there are any remaining issues, we would be grateful if you could discuss them with us during the discussion period. Thank you once again for your time and effort.

---

> > ### Comment · Reviewer_an6t · 2024-11-24
> >
> > Thank you to the authors for their time and effort in providing the detailed response. I also apologize for overlooking the experiments regarding preference feedback quantity in the original manuscript.
> >
> > Unfortunately, my main concerns regarding the evaluation remain:
> >
> > 1. While I understand the challenge of finding off-the-shelf baselines for this particular setting, the current evaluation (where the proposed method outperforms sparse-reward methods but underperforms more privileged methods) does not provide sufficient insight for readers to assess the effectiveness of DPM.
> > 2. The evaluation relies solely on StarCraft-based benchmarks, lacking diversity.
> > 3. The reward function ensembles introduce additional time and computational complexity during both training and evaluation, though this may be acceptable to some extent.
> >
> > Given these concerns, I will maintain my original rating. Thank you again for your response.

---

> ### Author Response · Authors · 2024-11-28
>
> Thank you for your valuable feedback. Please find our responses to the reviewer’s comments below:
>
> 1. **Baselines:** As mentioned earlier, our research faces challenges due to the lack of comparable preference-based MARL methods. However, we are currently attempting to modify MAPT, which was previously tested in an offline environment, to operate in an online setting to include it as a baseline. We will update the results as soon as they are ready.
>
>
> 2. **SMAC Environment only:** SMAC is one of the most well-known environments in MARL, and its diverse scenarios make it suitable for evaluating our method. Furthermore, unlike the dense-reward setting, SMAC with a sparse-reward setting remains a challenging task in the MARL domain. However, to support your evaluation of DPM, we are currently conducting additional experiments in environments such as GRF. Similarly, we will update the results as soon as they are available.
>
>
> 3. **Multiple reward functions:**  Using a single reward function can introduce bias. However, this can be mitigated by averaging the rewards generated by multiple reward functions. Additionally, we utilized an ensemble of reward models in the process of selecting pairs(or a transition) for preference feedback. Specifically, we calculated the Kendall's Tau value for the agents’ ranking generated by the reward models. Transitions $(s_t, \mathbf{o_t}, \mathbf{a_t})$ with the lowest agreement in these values were considered as data that the reward functions had not sufficiently encountered, and such transitions were selected for feedback. This approach is also employed in other PbRL methods. For a comparison with cases where this method was not used, please refer to Ablation (b) in Figure 8, which shows that performance significantly deteriorates when ensemble learning is not applied.

---

### Official Review · Reviewer_yPwC · 2024-11-03

**Soundness:** 3
**Presentation:** 2
**Contribution:** 2
**Rating:** 5
**Confidence:** 4

**Summary:**

This work introduces a preference learning method for the multi-agent RL setting. In particular, the approach, DPM, leverages traditional trajectory comparison feedback along with agent comparison preference feedback, which ranks agents' actions.

**Strengths:**

1. The results in SMAC and SMAC v2 show that adding DPM can significantly outperform other standard MARL baselines.

2. I liked how the authors included additional generalization experiments, demonstrating that DPM can work well with other MARL algorithms, not just QMIX.

**Weaknesses:**

1. Unclear what the contributions are / limited novelty:

This method's significant contribution seems to be its use of a new feedback methodology, where a teacher ranks specific actions.

2. Experimental Design:

The authors did compare DPM against several MARL baselines but not against the other preference learning algorithm for multi-agent systems [1]. I think the authors need to compare against this algorithm or explain why this would not be a valid comparison.

The evaluation is limited to one environment domain, SMAC. Why not include other prominent MARL environment suites, such as Multi Particle Environments (MPE) or Multi-Agent Mujoco?

3. Use of LLM as preferences:

The authors use LLMs to obtain preferences, which is a design choice. However, the reasoning for this design choice is limited. The authors note that DPM uses preferences from LLM to address the issues associated with human preferences. Can the authors elaborate on these issues? In addition, the authors should describe the limitations of using an LLM in this context.

We should also note that one primary objective in preference learning is to learn reward functions that encode human preferences (i.e., it is important that humans are actually providing the feedback). Can DPM work with human preferences? I’m a bit concerned that humans would have difficulty ranking specific actions in the agent comparison component of the approach. Especially in the continuous action setting, actions from one time step to the next are unlikely to change drastically, making it difficult for a human to detect differences.


[1] https://ojs.aaai.org/index.php/AAAI/article/view/29666

**Questions:**

1. How much preference feedback is being used in Figure 4? Is the same amount of the two types of preference feedback being used?

2. In the preference alignment case study, it is unclear what is being aligned. Can the authors elaborate on what is being compared?

---

> ### Author Response · Authors · 2024-11-20
>
> Dear reviewer, we sincerely appreciate your comprehensive feedback and constructive comments. We are also grateful for the opportunity to respond to your concerns. Our responses to the weaknesses and answers to the questions are presented below.
>
> ---
>
> ### **Weakness**
>
> **1. Novelty and contributions**
>
> (1) Novelty : Applying existing PbRL methods to MARL comes with inherent challenges. Due to the characteristic of multi-agent systems, where multiple actions occur in a single step and the space becomes significantly larger, comparing trajectories alone is insufficient for optimizing the reward function. In other words, in MARL, the temporal credit assignment problem, which involves difficulty in identifying key states (or actions) in trajectory comparison, becomes more pronounced. Our research overcomes these limitations by simply modifying the type of feedback used, a novel approach to addressing these challenges.
>
> Additionally, instead of using human feedback as in RLHF (Reinforcement Learning with Human Feedback), our research adopts AI feedback, aligning with the RLAIF (Reinforcement Learning with AI Feedback) perspective. This approach not only overcomes the limitations of RLHF but also provides motivation for further research in RLAIF.
>
> (2) Contributions : This is the first approach to combine PbRL with MARL in an online setting. While there has been extensive research on PbRL for single agents, there is very little research tailored to the multi-agent case. The only notable study, MAPT [1], relies on offline data to provide feedback and uses expert-level datasets (high win rates) with a very large number of feedbacks (over 10,000), making it unsuitable for online settings. In contrast, our study is the first PbRL research applied to MARL in an online setting. This not only demonstrates novelty but also establishes our research as a potential milestone and baseline for future PbRL+MARL studies.
>
> **2. Experimental design (fairness)**
>
> (1) Comparison with MAPT [1] : We hoped to compare PbRL methods with DPM. However, as mentioned in the Common Response, it was challenging to make comparisons due to the scarcity of MARL+PbRL researches. A fair comparison with MAPT, as referenced by the reviewer, is limited for the following reasons.
>
> - Differences in Setup
>   - MAPT is implemented from an offline RL perspective, whereas our study, DPM, applies to an online RL setting. As such, direct comparisons are not valid.
>   - The detailed differences between MAPT and DPM are as follows (presented in the table below)
>
> | Aspect                  | MAPT                        | DPM                          |
> |-------------------------|-----------------------------|------------------------------|
> | Setting                | Offline                     | Online                       |
> | Use of Expert Dataset  | True (Data generated from high-win-rate policies for feedback and policy learning) | False (Requires exploration, starting from scratch) |
> | Number of Preference Feedback | 50,000                   | 450–1,500                   |
> | Feedback Type          | Scripted Teacher (Ground truth reward) | LLM-Generated Feedback      |
>
> In summary, MAPT and DPM differ fundamentally in their setups (offline vs. online), resulting in significant differences, such as MAPT using far more feedback and higher-quality trajectories for generating feedback. Due to these differences, direct comparisons with DPM are not valid.
>
> (2) Reason for Selecting SMAC as the Main Environment
>
> There are two key reasons for selecting SMAC (StarCraft Multi-Agent Challenge) as the main environment:
>
> - Representative Environment for MARL
>
> First, SMAC is one of the most representative environments for MARL and offers scenarios covering a variety of cases. Experiments in SMAC not only provide a clear performance comparison with the baselines but also play a significant role in assisting MARL researchers in their evaluation. Therefore, we believed that experiments in diverse scenarios within SMAC would provide meaningful insights. Moreover, to address the environmental limitations of SMAC, we also conducted experiments in SMACv2 to enhance the evaluation and addresses the shortcomings of SMAC.
>
> - Limitations of DPM
>
> One of the limitations of DPM is that it requires ranking the actions of agents, meaning that the actions at a single time step must carry meaningful significance. However, in environments like Multi-Agent Mujoco, where it is difficult to evaluate the contribution of individual agents, applying this approach becomes challenging. Therefore, as mentioned in Future Directions(official comment by authors at 3.(3)), we are preparing future research on methods that calculate preferences by aggregating actions over a certain period.

---

> ### Author Response · Authors · 2024-11-20
>
> **3. Use of LLM as preferences**
>
> (1) Addressing Limitations of Human Feedback: As highlighted in RLAIF studies such as RLAIF[2], MOTIF[3], there are inherent limitations to relying on human feedback. When using human feedback, there are various challenges such as cost and consistency issues. Among these, the problem we aim to address using LLM is the inability to perform automatic and continuous learning in an online setting. This is because the learning process must pause to wait for human feedback. In contrast, DPM leverages LLMs as a source of feedback, enabling uninterrupted and continuous training until completion. This overcomes a critical bottleneck in online learning scenarios, where delays caused by waiting for human feedback can significantly hinder progress.
>
> (2) Compatibility with Human Feedback: While DPM utilizes LLM-generated feedback, it is also compatible with human feedback. The accompanying Figure 7 (and lines 466-479) demonstrates the performance of DPM when trained with human feedback. It shows that when both trajectory preferences and agent action preferences are used, DPM performs well. However, the absence of either type of feedback results in challenges during training. Details on the process of collecting human feedback can be found in the Appendix D.2 (lines 1019~1025).
>
> (3) Future Directions : In environments with continuous action spaces, feedback for single-step actions may not be sufficient. Instead, methods that aggregate actions over a few steps and provide feedback on these grouped agent actions could be more effective. Exploring this approach will be a focus of our future research.
>
> ---
>
> ### **Questions**
>
> **1. Amount of preference feedback**
>
> Is the question about the amount of preference feedback referring to Figure 5 rather than Figure 4? In Figure 5, the amount of feedback used for comparison is consistent across all cases. For example, in the case of DPM, a total of 150 feedback items were used, consisting of 75 trajectory comparison feedback and 75 agent action comparison feedback. Similarly, the 'Trajectory' baseline used 150 trajectory comparison feedback, and the 'Agents' baseline used 150 agent action comparison feedback. In other words, the same number of feedback items was used to ensure a fair comparison.
>
> **2. Explanation of the preference alignment case study**
>
> We have supplemented the explanation regarding this in the Appendix (line 859–900 and Figure 9). To summarize, we evaluate how closely the reward generated by the reward model, trained using various feedback types (trajectory only, agent only, and both (DPM)), aligns with actual human feedback, LLM, and ground truth-based scripted teacher feedback. For this evaluation, we use 100 trajectory pairs, not included in training, as the test set and collect feedback on these pairs (from human, LLM, and scripted teacher). When the reward model indicates a preference for trajectories with higher cumulative rewards, we measure how closely the generated feedback aligns with the actual feedback. As shown in Figure 9, DPM demonstrates better alignment across all three feedback sources, which correlates with its superior training performance.
>
> ---
> ### **References**
>
> [1] Zhu, Tianchen, et al. "Decoding Global Preferences: Temporal and Cooperative Dependency Modeling in Multi-Agent Preference-Based Reinforcement Learning." Proceedings of the AAAI Conference on Artificial Intelligence. Vol. 38. No. 15. 2024.
>
> [2] Lee, Harrison, et al. "Rlaif: Scaling reinforcement learning from human feedback with ai feedback." arXiv preprint arXiv:2309.00267 (2023).
>
> [3] Klissarov, Martin, et al. "Motif: Intrinsic motivation from artificial intelligence feedback." arXiv preprint arXiv:2310.00166 (2023).
>
> ---
> We hope our responses have addressed your concerns. If there are any remaining issues, we would be thankful and pleased to discuss them during the discussion period. Thank you once again for your time and invaluable feedback.

---

> ### Comment · Reviewer_yPwC · 2024-11-26
> **Response to Authors**
>
> Comparison with MAPT:
>
> - After reviewing MAPT again, it does not seem inherently offline. Trajectories are sampled from an offline dataset, but I believe it can be adjusted to sampling in an online fashion. In addition, the datasets are said to be extracted from the replay buffers of scripted teacher training [1], therefore I don't think they are solely expert trajectories.
>
> I will raise my score, but overall most of my concerns remain. As this is an empirical paper, the experiments need to be more thorough.
>
> [1] https://ojs.aaai.org/index.php/AAAI/article/view/29666

---

> ### Author Response · Authors · 2024-11-28
>
> First and foremost, thank you for your valuable feedback.
> As you mentioned, there is no explicit statement that MAPT [1] solely uses expert trajectories. However, the high performance of MAT mentioned in Table 1 of the MAPT paper (with rewards converging to the maximum value of 20) may imply that the quality of the replay buffer used by MAPT is high and at an expert level.
>
> Nonetheless, to support your evaluation of DPM, we are preparing experiments by enabling MAPT to operate in an online setting. Although the discussion period is limited, we will make every effort to complete the experiments within the time and upload the results. Once again, we sincerely appreciate your invaluable comments and efforts.
>
>
> [1] Zhu, Tianchen, et al. "Decoding Global Preferences: Temporal and Cooperative Dependency Modeling in Multi-Agent Preference-Based Reinforcement Learning." *Proceedings of the AAAI Conference on Artificial Intelligence*. Vol. 38. No. 15. 2024.

---

### Official Review · Reviewer_Jjs9 · 2024-11-05

**Soundness:** 4
**Presentation:** 3
**Contribution:** 3
**Rating:** 6
**Confidence:** 2

**Summary:**

In this work, the authors propose generating agent-based intrinsic reward functions in MARL whose functions are generated by combining a preference score based both on pairwise trajectory comparison as well as a comparison between agent decisions. The use an LLM approach to generate the preference comparisons and compare on both SMAC and SMACv2. The contribution is simple, but interesting and the paper explores many different comparisons which is well done.

The exposition and writing can be improved as well as small mistakes such as Figure 1. They compare against many baselines, but in a completely sparse reward setting where as the baselines were developed with a ”weakly” sparse reward which perhaps makes the comparisons unfair. Further, they have omitted some recent work in Sparse Reward MARL - see below. Additionally, only SMAC and SMACv2 are compared, but other MARL environments would be interesting to add - although, in this reviewer’s opinion are not necessary. Additionally, the time constraint of training through an LLM is only very briefly mentioned at the end and perhaps further analysis can be conducted on the interpretability of their learned reward functions compared to the dense human version.

**Strengths:**

*I strongly appreciate Figure 2.

*For feedback, they use an LLM for feedback of the RL agent. For completeness, the limitations of such a choice must be at least mentioned in the introduction, but I enjoyed the comparison with Human feedback in Appendix D.2 and preference Alignment.

*The paper structure overall is very clear and the idea is simple but intuitive.

*The ablation study is quite nice. I appreciated this as well.

**Weaknesses:**

* Figure 1b has no blue line or perhaps it is always zero of which it is useless and hard to see.
* Figure 1a is extremely confusing to follow.
* There is a real lack of related work section compared to what has been done in MARL on sparse versus dense reward functions. No details about the ideas the baselines have introduced in the sparse MARL space are discussed.
• What is Ψ in P_ψ, \hat{r}_ψ for example in equation (1) - I assume it is the learnable weights, but this should be stated.
* Probably should add that the loss is negative log-liklihood in Eqn. (2) as other approaches can be used
* In SMAC, it is very challenging to win with the completely sparse approach. Thus, I think a comparison with the ”less weak” sparse manner is more fair as I imagine the baselines will do better, but the results of DPM+QMIX will not change much. This ”less weak” sparse reward is very natural to construct of which a non-expert could reasonably create. Comparing for example with ”https://arxiv.org/pdf/2405.18110”, it is clear that for 5m v.s. 6m, the result is extremely similar in terms of performance if one uses their weakly sparse reward
* A comparison with the following sparse reward solution to SMAC in ICML 2023 should at least be mentioned or discussed. (”Lazy Agents: A New Perspective on Solving Sparse Reward Problem in Multi-agent Reinforcement Learning”)
* There are a few typos and the flow of the paper is incoherent at times. (End of Section 2.1 for example). Shortened List of typos:
– The title should be singular. "Dual Preference Based MARL"
– Section 2.1, second paragraph. Third sentence should be "in which subgoal-based..."
– Section 2.1, second paragraph. Fourth sentence should be "caused by"
– Section 2.2, 3rd, sentence "a scripted teacher which assigns"
– Fix the bibliography. Many things such as ”ai”, "Smacv2" etc. should be capitalized.
– Please use "LLM" as a singular vs plural correctly. For example in first pargarph of figure 3.1, one needs the article "an" in many places.
– Section 3.3., "common PbRL research"

**Questions:**

* The relative size of L^T versus L^A may not be the same as one is summing many more terms in (6). Is this addressed with some normalization?
* Is it fair to compare with sparse reward scenarios? I ask because essentially, the approach is building its own dense reward function via trajectory and agent preferences and so perhaps it should be compared from a dense perspective.
* The temporal component of training through an LLM is not discussed (somewhat discussed in the conclusion, but not explicitly). How does it compare the computation of a densely constructed rewards as one needs to evaluate a NN for each agents intrinsic reward. This may scale badly as the number of agents become large compared to human constructed reward functions although subsets may be used as mentioned.
* It would be interesting (although not necessary) to do an analysis comparing the reward generated via DPM vs the dense reward for the problem. Perhaps the DPM generated reward has some sense of interpretability.

---

> ### Author Response · Authors · 2024-11-20
>
> Dear reviewer, we appreciate your thorough review and valuable feedback. Our responses to your comments and questions are presented below.
>
> ---
>
> ### **Weakness**
> **1. Main text and figure revision**
>
> We kindly ask the reviewers to refer to the updated paper where the updated parts are highlighted in blue color.
> - Revision to Figure 1(a),(b) : lines 54-71
> - Additional explanation about $\psi$ and the loss function : lines 137-139
> - Addition to the related works : lines 108-116 and lines 783-809
> - Typographical error correction
>
> **2. Enhance related work (MARL algorithms in sparse reward environments)**
>
> Researches on MARL algorithms in sparse reward environments have been summarized in the background section (2.1), categorized by approach. Detailed descriptions of these algorithms and additional information on the referenced paper (LazyAgent) [1] have been added to the Appendix (lines 783–809). VDN, QMIX, QPLEX are researches under dense reward settings, while MASER, FOX, ICES and LAIES (LazyAgent) focus on sparse reward problems.
>
> **3. Additional experiments in weak reward settings and another mentioned baseline (LazyAgents)**
>
> Please refer to item 6 in the Common Response.
>
> ---
>
> ### **Question**
> **1. Loss normalization**
>
> For a single feedback, $L^T$ is computed only once, whereas $L^A$ is repeated $_nC_2$ times depending on the number of agents. Therefore, it is necessary to divide by $_nC_2$ to balance the ratio between $L^T$ and $L^A$. We have revised the equation and added an explanation regarding this process (lines 257-261, Equation (6)).
>
> **2. Fairness in comparison**
>
> Please refer to item 2 in the Common Response.
>
> **3. Temporal component of training**
>
> It is true that NN-based reward functions require relatively more time compared to human-designed rewards, as the reviewer mentioned. However, this is not an issue unique to DPM but rather a general challenge in preference-based RL. To address this, we have designed the reward function to be as simple and lightweight as possible. While it is true that calculating individual rewards increases computational cost as the number of agents grows, our model offers advantages in terms of performance improvement and the ability to achieve this with fewer feedback instances compared to using a team reward function.
>
> **4. Comparison with ground truth rewards (dense rewards)**
>
> Please refer to item 4 in the Common Response.
>
> The comparison between the ground truth reward and the rewards generated by trained reward function has been added to Appendix B.1 (line 843-900 and Figure 9). When calculating the Peason Correlation Coefficient(PCC) between the rewards, DPM(using two types of feedback) achieves a value of approximately 0.7, wherease other cases show relatively lower correlation. This indicates that DPM is more aligned with the groundtruth, resulting in better performance. This is consitent with the preference alignment case study (Figure 9(b)). As shown in the preference alignment results, DPM aligns better than the baselines across all sources, including human preference, LLM, and the scripted teacher.
>
> ---
>
> ### **References**
>
> [1] Liu, Boyin, et al. "Lazy agents: a new perspective on solving sparse reward problem in multi-agent reinforcement learning." International Conference on Machine Learning. PMLR, 2023.
>
> ---
>
> We hope our responses have addressed your concerns. If there are any remaining issues, we would be thankful if you could share them during the discussion period. Thank you once again for your time and invaluable input.

---

> > ### Comment · Reviewer_Jjs9 · 2024-11-27
> > **Response to author's rebuttal**
> >
> > Thank you for your changes. (There remains a small typo above eqn (2)) ie should be “is as follows:“.) I'll improved my score

---

> ### Author Response · Authors · 2024-11-28
>
> We sincerely thank you for your support and taking the time to review our paper. We have corrected the typos you mentioned.
>
> If you have any further unresolved questions, we would greatly appreciate it if you could share them with us.
>
> Best Regards,
>
> The Authors.

---

### Author Response · Authors · 2024-11-20
**Common Response**

### **5. Revision of Main Text and Figures**
Thank you for pointing out the typos. we have corrected the mentioned typos as well as other typos throughout the paper. Revisions have been highlighted in blue in the text.

### **6. Additional experiments in a weak sparse reward environment**
Reviewer Jjs9, an6t and YJrV raised questions about experiments under weak sparse reward setting. Many researches have adopted weak sparse reward settings and to support your evaluation of our approach, we conducted additional experiments comparing the performance of our method with the baselines in a weak sparse setting(Figure 10, lines 902-909). Since our method is not significantly influenced by extrinsic rewards, we evaluated it in a hard reward setting, and the results demonstrate that our method outperforms the baselines even in more extreme environments. Additionally, we compared our approach with the method mentioned in the paper reviewer Jjs9 referenced(LazyAgent). In the case of LazyAgent, additional information must be proivded, requiring human experts to anlyze observations and categorize them appropriately, making it difficult to consider this a completely fair comparison under identical conditions. Despite this, our method consistently shows better performance across various scenarios.

---
[1] Zhu, Tianchen, et al. "Decoding Global Preferences: Temporal and Cooperative Dependency Modeling in Multi-Agent Preference-Based Reinforcement Learning." Proceedings of the AAAI Conference on Artificial Intelligence. Vol. 38. No. 15. 2024.

[2] Klissarov, Martin, et al. "Motif: Intrinsic motivation from artificial intelligence feedback." arXiv preprint arXiv:2310.00166 (2023).

[3] Warnell, Garrett, et al. "Deep tamer: Interactive agent shaping in high-dimensional state spaces." Proceedings of the AAAI conference on artificial intelligence. Vol. 32. No. 1. 2018.

[4] Lee, Kimin, Laura Smith, and Pieter Abbeel. "Pebble: Feedback-efficient interactive reinforcement learning via relabeling experience and unsupervised pre-training." arXiv preprint arXiv:2106.05091 (2021).

---

### Author Response · Authors · 2024-11-20
**Common Response**

## **Common Response**
Dear reviewers, we would like to express our gratitude for your invaluable feedback on our paper. In response to the reviewers’ comments, we provide our common our answer below. We kindly ask the reviewers to refer to the updated manuscript.

---
### **1. Appropriateness of Experimental Setup (Fairness)**
Reviewer Jjs9, an6t, and YJrV raised concerns about whether it is fair to compare the proposed approach with sparse reward-based MARL algorithms. Since PbRL provides additional information in the form of preference feedback compared to general RL algorithms, it is preferable to compare PbRL algorithms against other PbRL approaches for a completely fair comparison. However, while PbRL has been extensively studied in single-agent RL, there is a lack of research in the MARL domain.
Although there is a related study (MAPT [1]) in the offline RL domain presented at AAAI2024, comparing it with our research would not be fair as it operates in an offline RL setting, utilizing expert trajectories and tens of thousands of preference feedbacks to train a reward model. Therefore, to establish baselines, we referenced the initial studies on PbRL and RLAIF to determine our setup.

MOTIF [2], as the first study to leverage LLMs for feedback, faced the challenge of having no comparable approaches and thus set RND, an exploration method known for sparse reward environments, as the baseline. Inspired by this, we set algorithms for sparse reward environments as our baseline. Furthermore, 'Trajectory' in Figure 5 can be considered as basic PbRL algorithm in single-agent RL.
Additionally, researches such as DeepTAMER [3] and PEBBLE [4] compared the performance of PbRL methods with general RL algorithms (e.g., SAC, PPO) in dense reward settings. Similarly, we used QMIX as the baseline in dense reward settings.

In summary, since there is no prior research applying PbRL to MARL, a perfect basline setting is limited. However, we conducted experiments to evaluate our method from various perspectives, and we believe these experiments will be helpful in evaluating DPM. Finally, we believe our study is a pioneering effort in PbRL research within MARL, laying the foundation for future research and serving as a milestone in this field.

### **2. Performance Comparison in Cases Using Human Preference**
Reviewer yPwC and YJrV raised concerns about whether DPM can utilize human feedback for learning. Of course, DPM is capable of learning based on human feedback. Figure 7 (lines 473-479) illustrates the performance of DPM when human feedback is used. As shown, DPM performs well when it utilizes both trajectory and agent action preferences, whereas it struggles in scenarios where this is not the case. The process of collecting human feedback is explained in detail in Appendix D.2 (lines 1015-1025).

### **3. Regarding Prior Knowledge (LLM)**
Reviewer NVua and YJrV raised questions about prior knowledge of LLMs. Large Language Models (LLMs) are trained on vast amounts of data during pretraining, making it highly likely that they possess prior knowledge about the problems we aim to solve, including SMAC. However, for models like GPT, where training data and details are not disclosed, it is impossible to verify whether specific prior knowledge has been learned. To address this, we conducted experiments by modifying the prompts through prompt engineering to minimize prior knowledge as much as possible. We conducted experiments by removing SMAC-related information from the prompt, as shown in Figure 12 (line 918-944), and the results are presented in Figure 13 (line 948-959). The LLM is able to generate proper feedback even in the absence of explicit SMAC-related information. Moreover, in the 2m_vs_1z scenario, the performance was better when SMAC information was excluded. This suggests that while the LLM may possess prior knowledge about SMAC, such knowledge does not always contribute positively to generating feedback.

### **4. Alignment with Extrinsic Rewards or Human Preference**
Reviewer yPwC and YJrV raised questions about alignment with extrinsic rewards or human preference. We calculated the Pearson correlation coefficient (PCC) to compare the estimated reward produced by the reward function with the ground truth reward, as shown in Figure 9 (lines 843–900). The PCC was highest at 0.7 when both types of feedback were used, while it was lowest when only trajectory comparison feedback was used. Additionally, to evaluate how well the model aligns with actual LLM feedback, human feedback, and scripted teacher feedback, we measured the accuracy using 100 test trajectory pairs. Similarly, the highest accuracy was observed when both types of trajectory comparison feedback were utilized.

---

### Meta-Review · Area_Chair_Q6wU · 2024-12-19

**Metareview:**

The reviewers acknowledged that the paper tackles an important problem setting of preference-based RL for multi-agent systems, and proposes an interesting approach that accounts for the comparison of individual agent actions along with the comparison of overall trajectory. However, the reviewers pointed out several weaknesses and shared concerns related to limited experimental evaluation, which lacks diverse environments and suitable baselines. We want to thank the authors for their detailed responses. Based on the raised concerns and follow-up discussions, unfortunately, the final decision is a rejection. Nevertheless, this is exciting and potentially impactful work, and we encourage the authors to incorporate the reviewers' feedback when preparing a future revision of the paper.

**Additional Comments On Reviewer Discussion:**

The reviewers pointed out several weaknesses and shared concerns related to limited experimental evaluation, which lacks diverse environments and suitable baselines. Even after the rebuttal phase, these concerns about the limited experimental evaluation remained. A majority of the reviewers support a rejection decision and agree that the paper is not yet ready for acceptance.

---

### Decision · Program_Chairs · 2025-01-22

Reject